# Significance of gene variants for the functional biogeography of the near-surface Atlantic Ocean microbiome

Leon Dlugosch[1], Anja Poehlein[2], Bernd Wemheuer[2], Birgit Pfeiffer[2], Thomas H. Badewien[1], Rolf Daniel [2] & Meinhard Simon [1,3✉]

Microbial communities are major drivers of global elemental cycles in the oceans due to their high abundance and enormous taxonomic and functional diversity. Recent studies assessed microbial taxonomic and functional biogeography in global oceans but microbial functional biogeography remains poorly studied. Here we show that in the near-surface Atlantic and Southern Ocean between 62°S and 47°N microbial communities exhibit distinct taxonomic and functional adaptations to regional environmental conditions. Richness and diversity showed maxima around 40° latitude and intermediate temperatures, especially in functional genes (KEGG-orthologues, KOs) and gene profiles. A cluster analysis yielded three clusters of KOs but five clusters of genes differing in the abundance of genes involved in nutrient and energy acquisition. Gene profiles showed much higher distance-decay rates than KO and taxonomic profiles. Biotic factors were identified as highly influential in explaining the observed patterns in the functional profiles, whereas temperature and biogeographic province mainly explained the observed taxonomic patterns. Our results thus indicate fine-tuned genetic adaptions of microbial communities to regional biotic and environmental conditions in the Atlantic and Southern Ocean.

[1] Institute for Chemistry and Biology of the Marine Environment, University of Oldenburg, Carl von Ossietzky Str. 9-11, D-26129 Oldenburg, Germany. [2] Department of Genomic and Applied Microbiology and Göttingen Genomics Laboratory, Institute of Microbiology and Genetics, Georg-August University of Göttingen, Grisebachstr. 8, D-37077 Göttingen, Germany. [3] Helmholtz Institute for Functional Marine Biodiversity at the University of Oldenburg (HIFMB), Ammerländer Heerstr. 231, D-26129 Oldenburg, Germany. ✉email: m.simon@icbm.de

  

On the basis of hydrography, nutrients, plankton, and boundary conditions, regions in ocean basins with similar properties have been classified into biogeographic provinces. This concept proved valuable in understanding environmental and biotic constraints on the distribution of many marine pelagic eukaryotic taxa[1,2]. As prokaryotes are major drivers of global elemental cycles in the oceans due to their high abundance and enormous taxonomic and functional diversity[3], attempts have been made to adopt this concept to microbial communities on global or ocean basin scales[4,5]. In fact, recent studies revealed global biogeographic taxonomic patterns of pelagic microbial communities[6–9] whereas reports on regional microbial biogeography of an ocean basin are still very scarce[10–12]. The diversity of functional prokaryotic features greatly exceeds the taxonomic diversity due to horizontal gene transfer (HGT) and rapid evolutionary adaptation[13–15]. Therefore, it is important to consider functional traits in microbial oceanographic studies. However, as functional classification, for example, KEGG-orthologues (KOs) are defined by their metabolic function, catalyzing a specific reaction and do not reflect the more specific properties of a given gene orthologue regarding its kinetic features and/or temperature range and optima. These more refined modulations of enzyme functions varying under different environmental conditions are important features of the fitness of the respective organism[15–17]. To distinguish between prokaryotic populations in large-scale environmental gradients, sequence variants of the same KO within one taxon may give detailed insight in adaption to different environmental and biotic conditions.

For a comprehensive understanding of biogeographic and latitudinal diversity patterns of microbial communities and their significance in the biogeochemical cycling of elements and matter, it is essential to include their functional traits. Temperature and other environmental variables have been shown to be important predictors for functional profiles and patterns of oceanic microbial communities[7,18]. The temperature was also identified as the main predictor of taxonomic diversity of microbial communities in latitudinal gradients. However, different relationships have been reported, ranging from highest diversity at intermediate temperatures around 15 °C[7,19,20] to maxima between 25 and 30 °C[8,21]. Although metagenomic studies shed light on different facets of functional traits of the open ocean microbiome including the adaptation to temperature[7,9], distinct biogeographic patterns of relevant functional genes are yet poorly studied and have not been assessed systematically in an ocean basin including the major biogeographic provinces. Refined analyses of functional biogeographic patterns on an ocean basin scale are critically important to better understand constraints and drivers of the functional biogeography of oceanic microbial communities and the niche occupation of their members.

In order to address these questions, we investigated the taxonomic and functional diversity of microbial communities in the near-surface Atlantic Ocean, including a section in the Southern Ocean, along a 13,000 km transect between 62°S and 47°N. Functional traits of the Atlantic Ocean microbiome (AOM) were assessed by metagenomics analyses on the basis of KO and non-redundant gene profiles, thus enabling an assessment of the general metabolic functions as well as their gene variants to examine biogeographic and ecotype adaptations. The results showed pronounced biogeographic patterns, specified in distinct biogeographic clusters of taxonomic, KO, and gene profiles. Random forest models indicated that biotic variables contributed most to the biogeographic structuring of KO and gene profiles whereas temperature and province explained most of structuring the taxonomic profiles. Further, the temperature was identified as important in the turnover of gene variants along the transect.

## Results and discussion

**Basic features of the AOM**. Twenty-two stations spanning nine biogeographic provinces and a temperature range from 1 to 28 °C were visited (Fig. 1a, b and Supplementary Table 1). Samples were collected at 20 m depth and the 0.2–3.0 μm-fraction was subjected to paired-end shotgun Illumina sequencing resulting in a total of 206 Gb with a sample mean of 8.9 ± 5.3 Gb (Supplementary Table S2). After assembly (total assembly length > 17.52 Gb), 12.18 Million gene sequences were predicted, and from these sequences, we reconstructed the AOM reference gene catalog (AOM-RGC) containing 7.75 Million non-redundant protein-coding sequences ("genes" hereafter) of which 55.2% were taxonomically classified (Fig. 1c). Genes from 18,923 genomes ("taxa" hereafter), across all domains of life, were identified by searching gene sequences against reference genomes available from NCBI and ProGenomes. The taxonomy of each gene is thus represented by the taxonomy of the closest matching genome. It is important to note that our approach of taxonomic gene identification by using reference genomes tends to overestimate the total number of taxa by assigning genes to different but closely related genomes or incomplete taxonomic annotation in reference genomes (e.g., metagenome-assembled or single amplified genomes). The largest proportion of classified genes (45.7%) affiliated to Bacteria whereas minor proportions to Archaea, viruses, and picoeukaryotes (together 9.5%; Fig. 1c). Thirty-eight percent of genes were functionally annotated by homology to a KO. In addition, 0.43% of genes were assigned to a CAZyme family (Fig. 1c). In total, 49.8% of high-quality reads were mapped to genes with known functionality. Fifty-nine percent of the AOM-RGC overlapped with the Tara Ocean Microbial RGC.v2[9] at 95% sequence identity; 12.7% of novel genes were functionally classified by similarity to a KO and 28.3% of novel genes remained functionally unclassified (Fig. 1d). Novel genes were predominantly detected outside the tropical Atlantic (SATL and WTRA) (Supplementary Table 2). This large fraction of novel gene sequences underline that despite recent efforts, we are still far from a comprehensive assessment and understanding of genomic features of epipelagic oceanic microbial communities and their participation in biogeochemical processes, especially in temperate and polar biomes. To estimate gene abundance of any given sample, high-quality paired Illumina reads were mapped onto the AOM-RGC, resulting in a mean yield of 76.7 ± 5.0% mapped reads. Accumulation curve analyses indicated that the AOM-RGC encompassed almost the entire richness regarding taxonomic and functional (KO) features and the great majority of genes (Supplementary Fig. S1). For the comparative analysis of the taxonomic, KO, and gene profiles, we only used genes that were taxonomically and functionally (KO or CAZyme) classified (41.1 ± 5.3% of all mapped reads) to keep the dataset consistent for analyses.

**Taxonomic and functional biogeography**. The nine biogeographic provinces[22] were characterized by distinct differences in temperature, salinity, annual mean concentrations of nitrate and phosphate, transparency, and chlorophyll *a* (Supplementary Table 1 and Supplementary Figs. 2 and 3). Bacterial biomass production did not show such biogeographic patterns (Supplementary Table S1 and Supplementary Fig. 3), presumably because it reflects the heterotrophic bulk activity of the entire prokaryotic communities which is more related to local substrate availabilities and spatio-temporal growth dynamics as a consequence of phytoplankton blooms and less to general biogeographic features[23]. In general, lineages of the major phylogenetic groups such as Cyanobacteria, Alphaproteobacteria, Gammaproteobacteria, Flavobacteria, Actinobacteria, Archaea, and picoeukaryotes exhibited

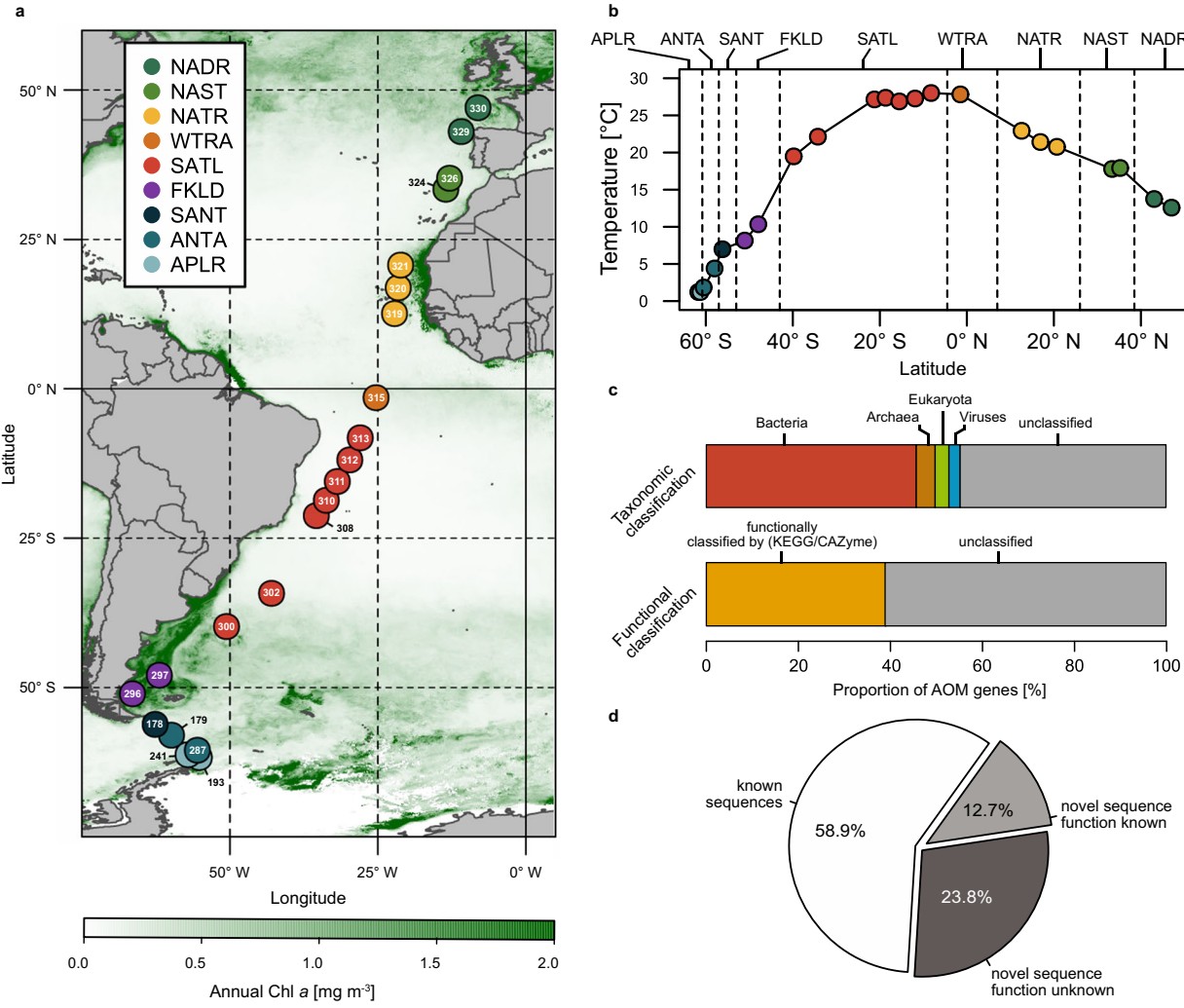

**Fig. 1 Stations in the Atlantic and the Southern Ocean visited during cruises ANTXXVIII/4 and -/5 with RV Polarstern for assessing the Atlantic Ocean Microbiome (AOM) and basic AOM features. a** Longhurstian provinces[22] of stations: Antarctic polar (APLR), Antarctic (ANTA), Subantarctic water ring (SANT), Southwest Atlantic Shelves (FKLD), South Atlantic gyre (SATL), Western tropical Atlantic (WTRA), North Atlantic gyre (NATR), North Atlantic subtropical (NAST), and North Atlantic drift (NADR). For further station details see Supplementary Table 1. Stations are overlayed on a map with annual mean concentrations of chlorophyll *a* at the surface (https://oceandata.sci.gsfc.nasa.gov). **b** Water temperature at 20 m sampling depth. **c** Percentages of taxonomic (Bacteria, Viruses, Archaea, Eukaryota, and unclassified) and functional (KEGG, CAZymes, and unclassified) annotations of 7.75 Million nr-gene sequences of the AOM. **d** Comparison of AOM gene novelty to the reference gene catalog of the Tara-Ocean data set (OM-RGC.v2).

different distribution patterns in the various provinces (Supplementary Fig. 4) and were mostly in line with previous reports on a global or comparable ocean basin scale[6,7,9,11,20].

Richness and diversity of the taxonomic, KO, and gene profiles showed systematic variation with latitude and ambient temperature (Fig. 2a–f). With the exception of KO richness, all profiles showed a bimodal latitudinal distribution with maxima around 40–50°S and 30–40°N and minima in the tropical region between 0 and 20°S and in higher latitudes beyond 50°S and 40°N. These patterns correspond with diversity and richness maxima at approximately 15–20 °C. In contrast to the KO and gene profiles, taxonomic diversity did not exhibit a well-defined peak but remained relatively constant above 10 °C. The richness and diversity patterns of the KO and gene profiles suggest that intermediate temperatures and seasonally fluctuating environmental and biotic conditions[24,25] may promote functional (micro) diversification of prokaryotic communities that are more reflected in community function than in taxonomic composition. Such conditions have been shown to favor HGT and diversification[14,17,26,27] and prevent microbes from eliminating genes, which may be discarded under more stable conditions leading to

more streamlined genomes[28]. Regions in the Atlantic Ocean with intermediate temperatures typically exhibit these pronounced seasonal hydrographic and/or biotic fluctuations[22,24,25] thus sustaining a greater functional diversity when compared to the permanently cold or stratified warm regions such as the Southern Ocean or oceanic subtropical gyres. Correlation analysis of richness and diversity (see methods section) and read count showed no correlation in the taxonomic and gene profiles (Pearson correlation, 0.22 and 0.27, $p > 0.05$) but were significantly correlated to KO richness (0.78, $p \leq 0.01$).

The taxonomic, KO, and gene profiles of the AOM were structured into distinct clusters, mostly in line with their respective biogeographic provinces or regions with similar hydrographic conditions (Fig. 2g–i, Supplementary Fig. 3). We determined optimal numbers of clusters by using their silhouette coefficient. All clusters showed significant differences (Kruskal–Wallis test, $p \leq 0.01$) of within-group distances to samples outside their respective cluster (Supplementary Fig. 5).

To identify environmental factors potentially driving the changes within each community profile, we fit random forest

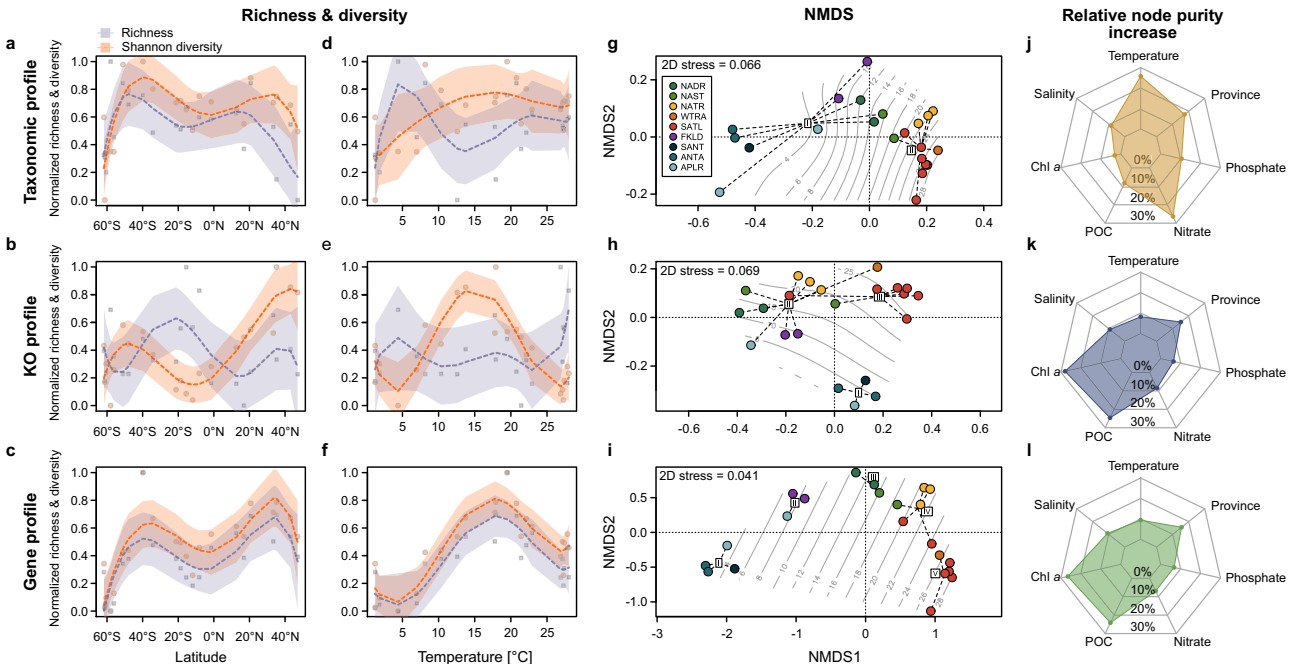

**Fig. 2 Taxonomic and functional diversity features of the AOM and related to in situ temperature. a–c** Normalized richness and diversity of taxonomic, KO, and gene profile of the AOM in relationship to latitude. **d–f** Relationship of normalized taxonomic, KO, and gene richness and diversity with temperature. Shading indicates the 95% confidence interval of the regression. **g–i** NMDS of Bray–Curtis dissimilarity of taxonomic, KO, and gene profiles. Isoclines approximate ambient temperature. Clusters were determined using Ward.D2 clustering and an optimal number of clusters was determined using the silhouette coefficient. **j–l** relative increase of node purity of environmental variables in observed clusters of taxonomic and functional profiles, determined by random forest models (number of trees = 500).

models based on hydrographic and biogeochemical properties of stations. The models explained 61% of the variance in the taxonomic profile and 36.5% and 35.8% of the variance in the KO and gene profiles, respectively. Taxonomic cluster (tC) I of the taxonomic profile encompasses stations from the Antarctic to subtropical regions whereas stations from tCII are located in the subtropics and tropics between 40°S and 34°N (Fig. 2g). Ambient temperature, annual mean nitrate concentration, and province affiliation were the most influential factors explaining the separation of the two clusters at approximately 18 °C and nitrate concentrations of 4 and 0.8 µM in the South and North Atlantic, respectively (Fig. 2j). This illustrates the compositional difference between the subtropical gyres and temperate and (sub)antarctic regions and the strong impact of temperature on the composition of the near-surface AOM, while biogeochemical and nutrient properties affected the cluster separation only to a lesser degree.

The KO profile showed separation into three distinct clusters (Fig. 2h). Functional cluster (fC) I comprise exclusively Antarctic and subantarctic stations (APLR, ANTA, and SANT) while fCII comprised stations from the northern, (NADR and NAST), central (NATR, WTRA), and austral temperate regions (FKLD) as well as station 193 (APLR) which is distinct from the other stations in the Southern Ocean due to limited mixing of water masses in the Bransfield-Straight with the Southern Ocean[29]. Functional cluster III encompasses all SATL stations as well as the southernmost NAST station. In contrast to the taxonomic profile, POC and chlorophyll-*a* concentration and not temperature or province was the most influential environmental factors separating the three functional clusters (Fig. 2k). The gene profile showed the highest differentiation into five clusters containing mostly adjacent stations from only a few oceanic provinces (Fig. 2i): Gene cluster (gC) I and gCII encompass stations of the Southern Ocean and south temperate Atlantic, gCIII stations of the temperate North Atlantic (NADR and NAST) and gCIV and

gCV stations of the (sub)tropical North Atlantic (NATR and NAST), WTRA and SATL. Similar to the KO profile, POC and chlorophyll-*a* were the most influential factors explaining the observed clustering of the profile (Fig. 2l). Overall variance explained was lower in gene and KO random forest models compared to the taxonomy. Temperature, as well as inorganic nutrients, contributed only little (<10%) to node purity in observed functional and gene clusters, while biotic features, i.e., biogeochemical and nutrient variables related to phytoplankton biomass (chlorophyll *a*, POC) and thus primary production, showed highest effects on cluster separation. These results suggest that community function is less determined directly by hydrographic properties but rather by complex interactions within the microbial communities, primary producers, and available nutrients. Our findings further imply that holistic ecological features, including the long-term environmental and biotic state as well as seasonal variability, can potentially better explain the large-scale structuring of oceanic microbial communities than single environmental variables. Although not entirely unexpected, our results emphasize the complex nature of niche segregation, community assembly, and functional profiles on a regional scale.

To explore functional differences between clusters of the KO profile, we performed a DESeq2 differential abundance analysis of KOs. Overall, we observed enrichment of 27.6% KOs and 27.4% CAZymes between fCI and fCII, 15.7% KOs, and 25.9% CAZymes between fCI and fCIII, and 14.7% KOs and 10.7% CAZymes between fCII and fCIII (Supplementary Fig. 6). For a more refined analysis, we focused on metabolic pathways and transport systems that are likely to be affected by the availability of inorganic nutrients and organic substrate (Fig. 3, Supplementary Data 1). Comparing fCI and fCII, the latter was enriched in genes encoding oligosaccharide, lipid, and amino acid transport, as well as in genes encoding nitrate uptake and reduction, denitrification, nitrogen fixation, and photosystems I and II

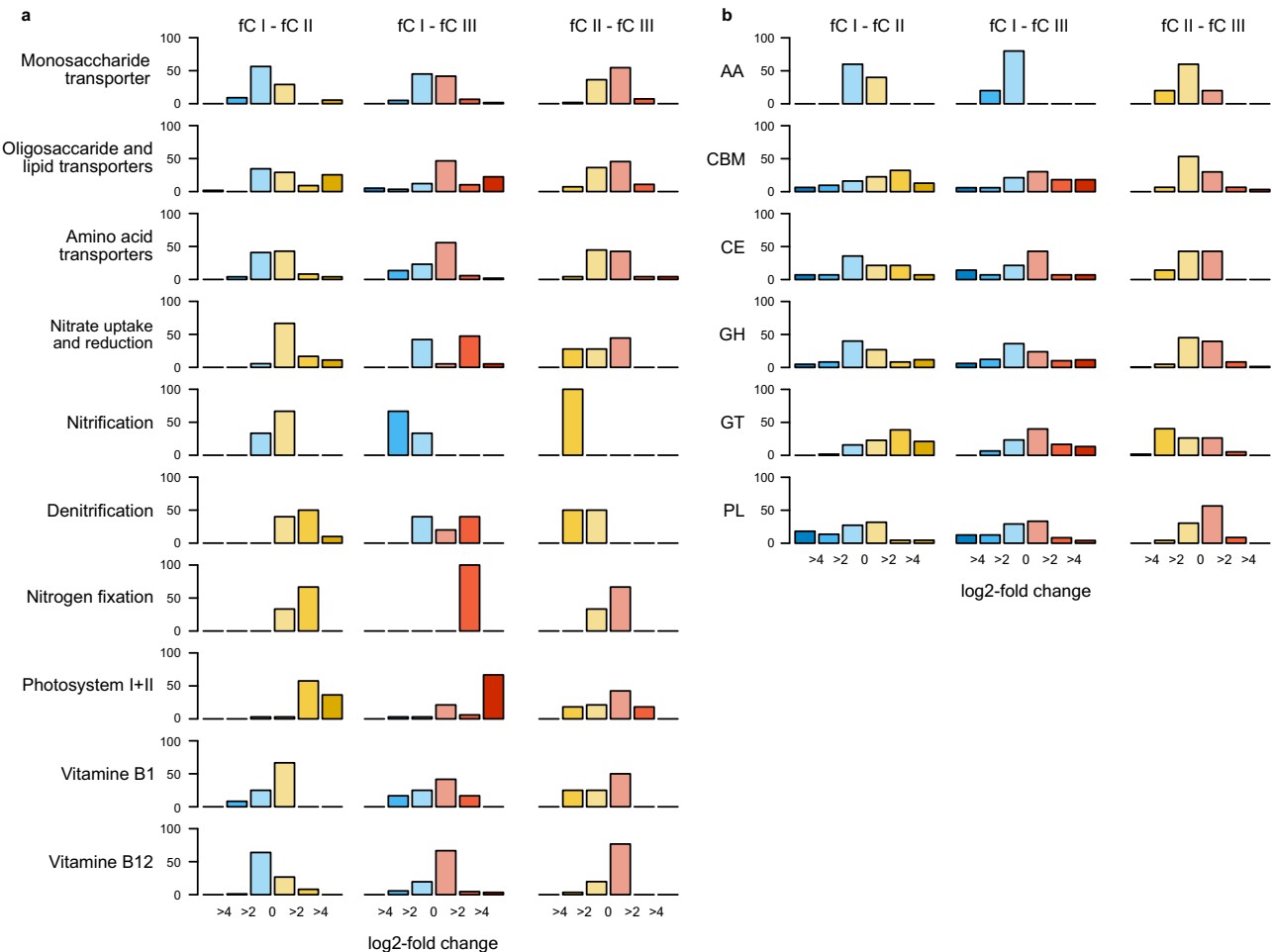

**Fig. 3 Differential KO abundance between functional clusters (fC) of the AOM. a** Proportion of pairwise differential KO abundance of selected pathways and functional groups between fC I (blue), fC II (yellow), and fC III (red) determined by DESeq2. **b** Pairwise differential CAZyme abundance between the fC I–III as in a (for abbreviations see legend of Fig. 4). Binned log2-fold change 0–2: no or weak difference between clusters, 2–4: intermediate enrichment, >4: strong enrichment.

(Fig. 3a). Similar enrichments of genetic features were also discernible when comparing fCI and fCIII but in addition, this comparison showed enrichment in fCI in genes encoding nitrification. The enrichments in fCII and fCIII of KOs encoding nitrate uptake and photosystems are likely reflecting the increased abundance of Cyanobacteria. Whereas fCII and fCIII exhibited only little differences in genes encoding substrate transport systems, fCII was enriched in genes encoding nitrification. Genes encoding synthesis of vitamins $B_1$ and $B_{12}$ were rather similarly distributed among the clusters.

CAZymes are likely to reflect biogeographic patterns of carbohydrate supply[30]. Therefore we assessed the biogeography of genes encoding the different families of these enzymes. Genes encoding glycosyltransferases (GT) and glycosylhydrolases (GH) largely dominated the CAZyme families (Fig. 4). Genes encoding GT, GH, and carbohydrate esterases (CE) were widely shared among many taxa whereas the carbohydrate-binding module (CBM), polysaccharide lyases (PL), and auxiliary activities (AA) were restricted to only a few lineages. Major players in carbohydrate metabolism were only a few lineages of Alphaproteobacteria, Gammaproteobacteria, and *Flavobacteriaceae* (Supplementary Fig. S7). In addition, Cyanobacteria were important players in carbohydrate metabolism, presumably reflecting the close link of carbohydrate metabolism to photosynthesis. To assess the biogeographic patterns of genes encoding the

degradation of carbohydrates, a major carbon source and prominent component of marine dissolved organic matter[31], we analyzed the distribution of genes encoding CAZymes in the three clusters of the functional profile. Our analysis shows pronounced enrichments of genes encoding various CAZyme families among functional clusters of the AOM (Fig. 3b), suggesting pronounced differences in available substrates among the biogeographic provinces. Especially fCI showed a substantially lower abundance of the CAZyme families CBM, CE, GT, and GT and enrichment of PLs. Differences between fCII and fCIII were mostly restricted to the CE and GT families. CAZymes with auxiliary activities were distributed evenly among the three clusters.

As carbohydrate turnover is an important biogeochemical process in all oceanic regions with greatly varying temperatures we further tested the relationship of the genes encoding CAZymes to temperature. Thirty-one percent of these genes across all CAZyme families exhibited significant correlations to temperature but those of GH and GT were most prominent (Fig. 4a). CAZyme genes were affiliated to three distinct clusters with different relationships to temperature (Fig. 4b, Supplementary Fig. 7). CAZyme-Cluster I exhibited an optimum at low temperatures, cluster II at intermediate temperatures around 15 °C, and cluster III at high temperatures. A taxonomical breakdown of CAZyme abundance shows that carbohydrates are

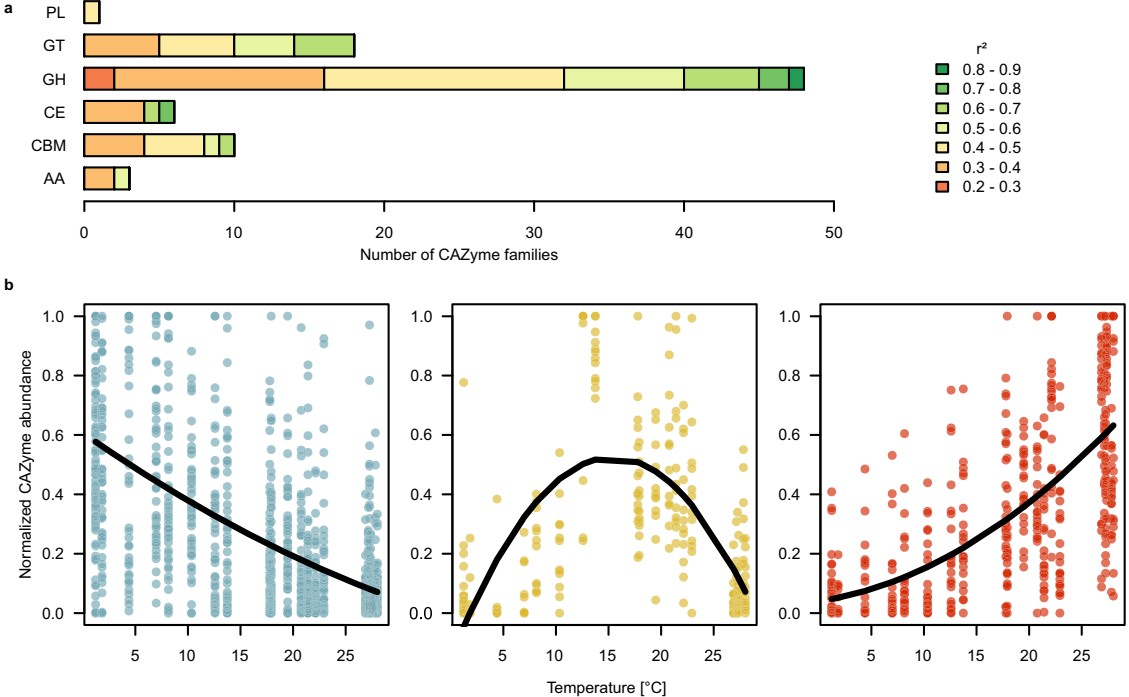

**Fig. 4 Temperature-dependent distribution of CAZyme families in the AOM. a** CAZyme families (AA auxiliary activities, CBM carbohydrate-binding modules, CE carbohydrate esterases, GH glycoside hydrolases, GT glycoside transferases, PL polysaccharide lyases) with normalized abundance profiles with a significant ($p \leq 0.05$) relationship to temperature, determined by unimodal regression models. **b** Relationships between temperature and CAZyme-family cluster abundance profiles determined by unimodal regression analysis. Cluster 1 ($n = 51$): $r^2 = 0.37$, $p < 0.001$; Cluster 2 ($n = 12$): $r^2 = 0.43$, $p < 0.001$; Cluster 3 ($n = 23$): $r^2 = 0.47$, $p < 0.001$.

metabolized very differently and by different phylogenetic lineages in the various biogeographic regions, from the nutrient- and diatom-rich temperate and subpolar regions to the nutrient-depleted and Cyanobacteria-dominated tropics (Supplementary Fig. 8). In addition to previously known carbohydrates utilizing *Flavobacteriaceae* and certain lineages of Gammaproteobacteria, our analysis identified Pelagibacterales, *Rhodobacteraceae*, and *Sphingomonadaceae* as important players in carbohydrate utilization. Such a detailed analysis provides refined insight into biogeographic patterns of marine microbial communities metabolizing carbohydrates, a major carbon, and energy source in the oceans.

These analyses of genes encoding KOs and CAZymes of the Southern and Atlantic Ocean provide detailed insights into regional differences in gene abundances, especially those involved in resource acquisition, despite an overall high functional redundancy of the KO profiles.

**Distance decay relationships and temperature-limitation of gene variants.** Distance–decay relationships have been used to determine boundary conditions of species dispersal or selection across large environmental gradients[7,20]. These relationships can also be used to assess taxonomic and functional turnover along such gradients. As enzyme function and stability are likely to be affected by temperature[17], we applied this approach to investigate community structure and function over geographic distance and the complete temperature range along the transect. Our results show that gene profiles exhibit a dissimilarity of >50% already at distances of <500 km and at temperature differences of 2–4 °C whereas taxonomic and KO features remained much more similar (Fig. 5). At greater temperature differences and geographic distance, the dissimilarity of gene profiles increased much

more than taxonomic and KO features and reached dissimilarities of >80% at distances of 2500 km and temperature difference of 7 °C. At distances of >5000 km and temperature differences of >15 °C dissimilarities reached 90%. In contrast, taxonomic dissimilarity remained <60% even at the largest temperature difference and <50% at the greatest distances and KO dissimilarities were always <20%, implying high KO redundancy, as reported previously[7]. This analysis shows a high turnover or modification of the gene profiles within the prokaryotic communities and a relatively stable overall community function in the Southern and Atlantic Ocean. Although a given KO can be ubiquitously present in the ocean, genes in closely related organisms may differ, probably to adapt to regional environmental conditions[17]. This eventually leads to sub-species functional variation and, over time, promotes the growth of specific regional populations carrying more efficient gene variants. Recent findings of high gene variant turnover of microbial communities over time at one location in the Mediterranean Sea[16] and in the global oceans[9] are in line with this notion. Diversification and adaptation of microbes to new ecological niches operate via mutation, HGT, dispersal, and invasion and are affected by population size[16,17,26,32]. This may lead to gaining completely new functional traits by HGT but also to modifying functional traits by mutation such as altered temperature and concentration ranges, substrate affinities, and optima of nutrient uptake systems as well as other metabolic properties[33–35].

To gain further information on the effect of temperature on gene turnover we examined the abundance of gene variants of single KOs originating from single taxa, i.e., genome-level annotation (see methods section), and their relationship to temperature. These gene variants showed distinct maxima and minima along the transect which often coincided with boundaries between adjacent biogeographic provinces and especially between

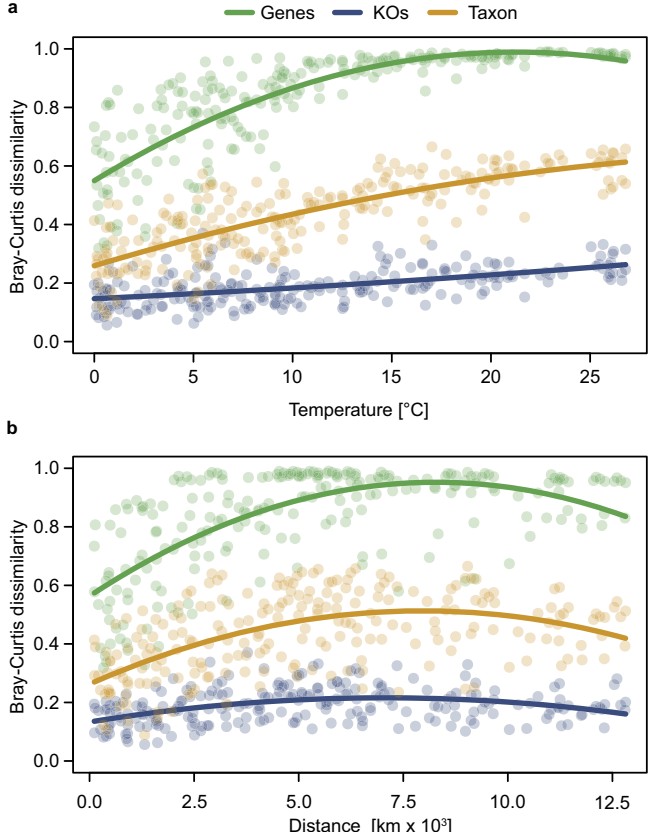

**Fig. 5 Bray–Curtis dissimilarity relationships of species, KO, and nr gene profiles of the AOM for distance and temperature. a** Relationship between Bray-Curtis dissimilarity of taxonomic ($r^2 = 0.65$, $p < 0.001$), KO ($r^2 = 0.29$, $p < 0.001$) and gene ($r^2 = 0.69$, $p < 0.001$) profiles to temperature difference. **b** Geographical distance of taxon profile: $r^2 = 0.30$, $p < 0.001$; KO profile: $r^2 = 0.15$, $p < 0.001$; gene profile: $r^2 = 0.45$, $p < 0.001$).

subpolar, temperate, and equatorial regions. Some of these variations occurred in regions with similar hydrographic properties in both hemispheres (Supplementary Fig. 9). The mean temperature range of abundant taxa and their gene variants (≥15% of maximal taxon/variant abundance) varied substantially among taxa and the mean temperature range was $8.6 \pm 3.5\,°C$ (Fig. 6a, Supplementary Data 2 and Supplementary Fig. 10). Several species occurred at higher or lower ambient temperatures with a rather limited temperature range such as *Prochlorococcus*, *Synechococcus*, SAR92, and *Planktomarina temperata*. Others and in particular members of the *Pelagibacteraceae* exhibited a large temperature range, reflecting their occurrence over large areas of the transect. Taxa of this family, e.g., *Cand*. Pelagibacter, HIMB5, harbored KOs with gene variants ranging from 1.5 to 26.3 °C. In contrast, KOs of other taxa exhibited gene variants with a much lower temperature range (<12 °C), such as the Gammaproteobacteria TMED* and Cellvibrionales bacterium TMED122, *Synechococcus* and *Prochlorococcus* (Supplementary Data 2). Cyanobacteria and Archaea generally exhibited relatively small temperature ranges, indicating a faster variant turnover and temperature adaptation. With few exceptions, ranges of gene variants were substantially lower than the overall temperature range of the respective taxon but increased with a higher overall taxon temperature range (Fig. 6a). This temperature adaption may partially explain the observed richness and diversity maxima in intermediate temperatures in the AOM gene profile. A larger number of variants can potentially coexist at these conditions

without detrimental loss of enzyme function, whereas enzymes adapted to very cold or warm temperatures can pose a substantial competitive disadvantage at both ends of the temperature gradient.

Based on this global analysis of the differential taxon gene turnover with temperature, we hypothesize that distinct KOs vary in their temperature adaptation and exhibit different temperature ranges. To identify such adaptations of KOs to temperature, we assessed the overall temperature ranges and abundances of gene variants of all KOs included in the functional AOM profile using KEGG BRITE classifications. The results show that different KOs vary in their temperature ranges and means (Fig. 6b). Although these differences are not statistically significant, some trends are worth exploring: Genes of pathways/functions directly involved in resource acquisition such as membrane transport, peptidases, replication, and repair exhibited the smallest temperature range while genes encoding energy metabolism, translation, energy- and carbohydrate metabolism exhibited larger temperature ranges. These data suggest a differential evolutionary pressure on functions towards temperature adaptation in broad functional categories. Distinct distribution patterns of gene variants over varying temperature ranges reflect different well-adapted populations of very closely related organisms. Biogeographic and seasonal dynamics of population microdiversity have been reported for prokaryotes as well as microeukaryotes as a reaction to different hydrographic conditions and environmental change[36–38]. These reports, however, looked at differently distributed KOs and ribotypes whereas our findings focused on functional genes enabling prokaryotic populations to occupy even more specific niches. Although our analysis was constrained to abundant taxa and orthologues, it is reasonable to assume that ecotypes with different variants of other genes can be found for taxa of most phylogenetic groups. Our results are consistent with the theoretical framework on niche occupation and sub-species diversification provided by Larkin et al.[17] and show the establishment of microbial communities consisting of taxa with presumably environmentally well-adapted ecotypes with similar metabolic traits, reflected by gene variants.

Genomic analyses have shown that closely related bacterial species harbor a large functional diversity with a relatively small core-genome and a much greater pan-genome, such as *E. coli* and *Shigella* spp.[39] or *Prochlorococcus*[36,40]. Considering microbial communities with an increasing number of taxa this discrepancy between taxonomy and functional traits would expand accordingly. Therefore, it may not appear surprising that we found a much higher dissimilarity with distance and temperature differences of the gene profiles relative to the taxonomic profiles and the distinct latitudinal patterns of gene variants. The great functional redundancy based on KOs may also appear to be expected considering that basic ecological processes like the fixation of $CO_2$ by phytoplankton primary production and decomposition of organic matter and cycling of elements by heterotrophic prokaryotic communities follow rather similar ecological principles in pelagic marine ecosystems irrespective of temperature and nutrient constraints. For a more refined understanding of the concerted and fine-tuned functioning of microbial communities and the adaptation to the given biotic and environmental conditions, it is most important to demonstrate these adaptive functional features on the level of gene variants and gene profiles, going beyond highly resolved taxonomic marker gene features[38,41]. The continuous exposure to the varying environmental and biotic conditions leads to well-functioning microbial communities and the continuous adaptation of their members, which in turn shape the observed latitudinal and temperature-related patterns of gene profiles. These functional features manifest on the sub-species and ecotype

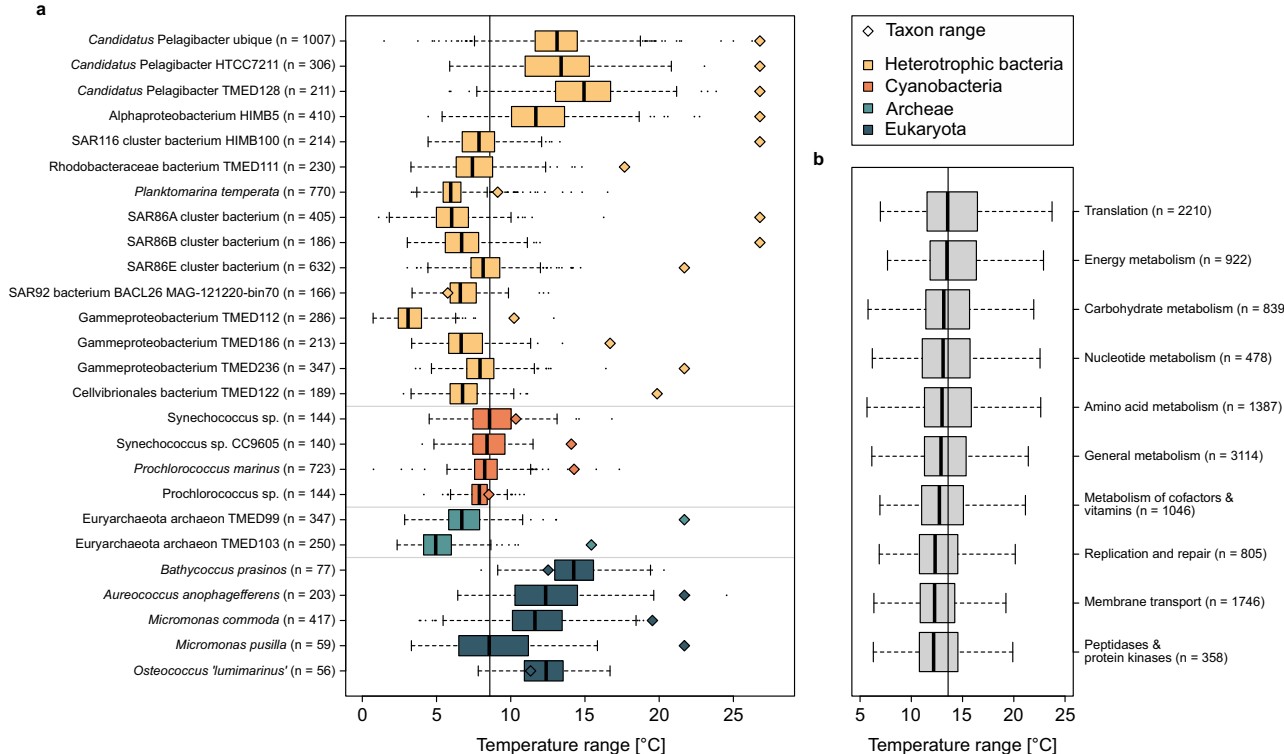

**Fig. 6 Mean temperature range of gene variants of highly abundant taxa and functional gene categories of the AOM. a** Box–Whisker plots of the mean temperature range of the abundance of single gene variants of all KOs in various abundant taxa of the AOM. The diamond indicates the temperature range of the taxon as a reference. **b** Box–Whisker plots of the mean temperature range of the abundance of gene variants of the listed KEGG BRITE. Vertical lines in each plot indicate the mean temperature range of all genes of the AOM. Numbers in brackets indicate the number of genes included in the analysis. Median values are represented by vertical lines, interquartile ranges shown as boxes, whiskers extending up to 1.5 times the interquartile range, and points showing outliers.

level and are not accounted for by analyzing KOs or taxonomic profiles alone. To expand our approach, it would be desirable to have a database that sufficiently resolves strain-level variation of KO variants.

The global oceans are subject to changing climatic conditions, which also affect the residing microbial communities and their continuous adaptation to ambient environmental conditions. Our analysis showed distinct richness and diversity maxima in taxonomic and functional community profiles at mid-latitudes and intermediate temperatures between 15 and 20 °C. Despite far-reaching functional redundancy in the AOM, we were able to identify differences among clusters of functional profiles in subpolar, temperate/subtropical, and tropical regions of the Southern and Atlantic Ocean. Although the temperature was identified as the most important predictor for the biogeography of community composition, community functions, and their biogeographic patterns were mainly explained by biotic variables related to primary production and the availability of organic matter and possibly other interactions. Differences between functional clusters were most pronounced in substrate transport systems as well as energy metabolism. A refined analysis of gene variants of a variety of metabolic pathways showed high turnover and relatively small temperature ranges of single genes, indicating a fine-tuned niche adaptation and occupation of members of the AOM. The functional adaption of prokaryotic communities to environmental and biotic conditions is evident on the level of gene variants and this genetic microdiversity impacts large-scale biogeographic microbial patterns and biogeochemical processes of the Atlantic and the Southern Ocean and presumably in other oceans and ecosystems as well.

## Methods
Twenty-two stations between 62°S and 47°N were visited during cruises ANT XXVIII/4, 13 March–9 April 2012, and ANT XXVIII/5, 10 April–15 May 2012, with RV Polarstern. For exact locations of the stations see table S1. Samples were collected at 20 m depth with 12 liter-Niskin bottles mounted on a Sea-Bird Electronics SBE 32 Carousel Water Sampler equipped with a temperature, salinity, depth probe (SBE 911 plus probe), a chlorophyll fluorometer (Wet Labs ECO—AFL/FL), and transmissometer (Wet Labs C-Star). For the analysis of particulate organic carbon (POC), total particulate N (TPN) and Chlorophyll a (Chl a) analyses, 1–4 l of seawater were filtered through Whatman GF/F filters and stored at −20 °C until analysis on board (Chl a) or in the home lab (POC, TPN). POC and TPN were analyzed as described previously[42], Chl a concentrations after extraction with 90% acetone for 2 h at −20 °C in the dark using a Turner fluorometer[43,44] calibrated with a standard chlorophyll solution (Sigma, St. Louis, MO). Bacterial biomass production was measured by the incorporation of ¹⁴C-leucine as described previously[45]. For metagenomics analysis, the water of several bottles was pooled in an ethanol-rinsed polyethylene barrel to a total volume of 40 l. Within 60 min after collection, the sample was prefiltered through a 10-μm nylon net and a filter sandwich consisting of a glass fiber filter (47 mm diameter, Whatman GF/D; Whatman, Maidstone, UK) and 3.0-μm polycarbonate filter (47 mm diameter, Nuclepore; Whatman). Picoplankton was harvested on a filter sandwich consisting of a glass fiber filter (47 mm diameter, Whatman GF/F; Whatman) and 0.2-μm polycarbonate filter (47 mm diameter, Nuclepore; Whatman). All filters were immediately frozen in liquid N and stored at −80 °C until further processing. Environmental DNA was extracted from the filter sandwich and subsequently purified employing the peqGOLD gel extraction kit (Peqlab, Erlangen, Germany) as described previously[46,47]. Illumina shotgun libraries were prepared using the Nextera DNA Sample Preparation kit as recommended by the manufacturer (Illumina, San Diego, USA). To assess the quality and size of the libraries, samples were run on an Agilent Bioanalyzer 2100 using an Agilent High Sensitivity DNA kit as recommended by the manufacturer (Agilent Technologies, Waldbronn, Germany). Concentrations of the libraries were determined using the Qubit® dsDNA HS Assay Kit as recommended by the manufacturer (Life Technologies GmbH, Darmstadt, Germany). Sequencing was performed by using the HiSeq2500 instrument (Illumina Inc., San Diego, USA) using the HiSeq Rapid PE Cluster Kit v2 for cluster generation and the HiSeq Raid SBS Kit (500 cycles) for sequencing in the paired-end mode and running 2 × 250 cycles.

**Metagenomic assembly and gene prediction**. Illumina reads were quality checked and low-quality regions, as well as adaptor sequences, were trimmed using Trimmomatic 0.36[48] (*ADAPTER:2:30:10 SLIDINGWINDOW:4:25 MINLEN:100*). The high-quality (HQ) reads were assembled using metaSPAdes 3.11.1[49,50]. Contigs smaller than 210 bp and average coverage <2 were discarded. Gene-coding sequences of the assembled contigs were predicted using Prodigal 2.6.2 in meta-mode[51]. Genes shorter than 210 bp and longer than 4500 bp were discarded to account for prokaryotic and eukaryotic gene length. This resulted in 12.05 Million unique gene sequences. To generate a gene catalog, gene sequences were clustered at 95% identity using USEARCH 10.0.24[52] (-cluster_fast –id 0.95). The resulting 7.75 Million cluster centroids were used as representative nr gene sequences. Sequencing and assembly statistics are summarized in Table S2.

**Taxonomic and functional annotation of gene-clusters**. Nonredundant gene sequences were taxonomically classified using Kaiju 1.6[53] (-greedy mode with 5 allowed substitutions and e-value 10e−5) with the Refseq nr (May 2018) and ProGenomes[54] database including prokaryotic, eukaryotic, and viral sequences. AOM sequence taxonomy represents the closest matching genome. Gene functions were assigned to AOM sequences using the Kyoto Encyclopedia of Genes and Genomes (KEGG) online annotation tool GhostKOALA[55] (https://www.kegg.jp/ghostkoala/) using the prokaryotic, eukaryotic and viral KEGG gene database (release 86) and default settings. In addition, AOM sequences were translated to amino acid sequence subsequently searched against the CAZy database (version: 2018-07-31) using DIAMOND[56] 0.9.30.131 blastx (-more-sensitive mode, cutoff e-value 10e-10 and ≥70% identity) to identify CAZymes[57]. To check for redundancy with genes of the Tara Ocean data set, sequences of the AOM-GC were searched against the Tara-Ocean OM-RGC.v2[9] using BLASTN (cutoff e-value 10e−10 and ≥95% sequence identity).

**Read abundance and normalization**. To acquire gene abundance data, HQ Illumina reads longer than 75 bp were mapped to the AOM sequences using bowtie2[58] 2.3.5 (-very-sensitive-local mode). Only the highest-scoring alignments were kept. SAMtools[59] version 1.9–58-gbd1a409 was used to convert the SAM alignment file to read abundance tables. Reads that did not map to any AOM sequence were discarded. To account for different sequencing depth and gene length, counts from each station were normalized by dividing read counts by gene length in kb to obtain reads per kilobase (RPK). Subsequently, scaling factors were calculated for each sample by dividing the sum of RPKs by one Million. The scaling factors were used to normalize the RPK values of each sample to counts per million (CPM)[60].

**Determination of marine biogeographic provinces**. Marine biogeographic provinces were determined by performing a cluster analysis (Euclidean distance, wards minimal variance criterion) stations according to their geographic position, temperature, salinity, and Chl *a* concentration as well as the descriptions in Sunagawa et al.[22]. Note that the position and extent of oceanic provinces underlie seasonal variation[61].

**Annual mean nitrate and phosphate concentrations**. Annual mean nitrate and phosphate concentrations at 20 m depth of each station were extracted from the 1° World Ocean Atlas 2018, provided by the National Oceanic and Atmospheric Administration (https://www.ncei.noaa.gov/access/world-ocean-atlas-2018/).

**Statistical analysis**. All statistical evaluations were performed in R (version 3.6.0; https://www.r-project.org/) using the additional packages vegan[62] (v2.5–6), ape[63] (v5.3), factoexrta[64] (v1.0.7), randomForest[65] (v4.6–14), DESeq2[66] (v1.30.1) and rtk[67] (v0.2.5.8).

**Richness and diversity**. To account for varying sequencing depth and unclassified genes, only taxonomically and functionally classified genes were used for the analysis of richness and Shannon diversity. Gene count tables were rarefed to 2 Million reads 99 times (Fig. S1) and subsequently, mean richness and Shannon entropy were calculated for nr gene-, KO, and taxonomic profiles (see above).

**Bray–Curtis distance of taxonomic/functional community profile and differential abundance analysis**. For the calculation of Bray–Curtis-dissimilarities between samples functional and taxonomically classified nr sequences of unrarefied abundance data were used. Taxonomic and functional datasets were summarized by using taxonomic classification on species/genome level and KOs, respectively. Samples were clustered using the Ward.D2 clustering algorithm. An optimal number of clusters was determined using the Silhouette-coefficient and validated by testing in-cluster vs. outgroup Bray–Curtis dissimilarities using non-parametric Kruskal–Wallis tests. Random Forest models (ntree = 500) using temperature, salinity, province as well as Chl *a*, POC, mean annual nitrate, and phosphate concentration (Supplementary Table 1) were fitted to identify the most influential environmental factors that can explain the observed clusters. Missing values in Chl *a* (*n* = 1), POC concentration (*n* = 2), annual mean nitrate (*n* = 4), and phosphate (*n* = 2) concentration were linearly interpolated (Supplementary Table 1). Resulting dissimilarities, geographic distance as well as the temperature difference between stations were used in linear models to determine the influence of temperature and geographic distance on taxonomic and functional community composition. Differential abundances of KOs and CAZymes between clusters of the KO profile (see above) were determined using DESeq2.

**Temperature range of gene variants**. The difference between minimal and maximal ambient temperatures with ≥15% of maximal gene variant abundance per taxon (Genome level classification, see above) was defined as temperature range. Only KOs with ≥10 variants per taxon were considered in this analysis.

**Distribution of CAZymes**. Abundance data from genes classified as CAZymes were normalized to values between 0 and 1. Unimodal regression model fitting was used to determine a relationship between temperature and family abundance, Benjamini and Hochberg[68] adjusted *p*-values ≤ 0.05 were considered significant. Euclidean distances of CAZy family abundance profiles significantly related to temperature were calculated and subsequently clustered using complete-linkage clustering. The same modeling approach was used to determine the relationship of clusters to temperature.

**Reporting summary**. Further information on research design is available in the Nature Research Reporting Summary linked to this article.

## Data availability

Sequence data generated in this study have been deposited in the European Nucleotide Archive[69] (ENA) under the INSDC accession number PRJEB34453 using the data brokerage service of the German Federation for Biological Data[70] (GFBio), in compliance with the Minimal Information about any (X) Sequence (MIxS) standard[71]. Environmental data from the cruise are available in the supplement and on PANGEA under the accession number PANGAEA.906247. The Atlantic Ocean Reference Gene Catalog (AOM-RGC), assembled contigs, and predicted genes are available at https://service.icbm.uni-oldenburg.de/data/AOM_data/.

## Code availability

The assembly pipeline, as well as scripts used for dataset generation and analysis, are available at https://github.com/LeonDlugosch/Atlantic-Ocean-Metagenomes.

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

## Acknowledgements

We thank the master, his crew, and the principal scientists (M. Lucassen, K. Bumke) of cruises ANT XXVIII/4 and -/5 of RV Polarstern, A. Gavrilov, H.-A. Giebel, S. Rack-ebrandt, T. Remke, J. Vollmers, M. Wietz, I. Wagner-Döbler, and M. Wurst for cruise support, M. Heinemann, B. Kuerzel, and R. Weinert for technical laboratory assistance.

This work was funded by Deutsche Forschungsgemeinschaft within the Collaborative Research Center *Roseobacter* (TRR 51) and was carried out in the framework of the Ph.D. Graduate Research training group "The Ecology of Molecules" (EcoMol) supported by the Lower Saxony Ministry for Science and Culture.

## Author contributions

L.D. carried out the bioinformatics and statistical analyses and wrote the draft of the publication; A.P. and B.P. carried out the metagenomics sequencing and quality control of the raw sequences; B.W. carried out sampling and sample filtration; T.H.B. carried out the hydrographic analyses and water mass identification; R.D. supervised the metagenomics sequencing and contributed to reviewing the paper; M.S. designed the study, supervised the bioinformatics and biogeographic analyses and finalized the draft paper. All authors reviewed the paper.

## Competing interests

The authors declare no competing interests.
