## [Peer Review File · Nature Communications]

Reviewers' Comments:

Reviewer #1:

Remarks to the Author:

This paper explores an incredible data set of metagenomes from mostly bacteria along a 13,000 km transect from the South Atlantic (62 S) to the North Atlantic (47 N). Although they discuss briefly other data, such as chlorophyll and nutrients, they focus on temperature and distance to explore how bacterial communities varied along the transect. As stated in the last sentence of the Abstract, the authors' main conclusion is that "gene variants," not KEGG-orthologues, "shape the functional biogeography of the Atlantic microbiome."

I think the authors can come up with a better conclusion and a more novel "take-home" message. First, although Figure 3 clearly shows differences among communities defined by gene variants, Figure 2, the other main argument for these differences, is not as convincing, as argued below in more detail.

Second, even if the authors are correct, which I'm sure they are, I doubt many readers would be surprised to hear that differences between communities were more evident among gene variants than among KEGG-orthologues. KEGG is broad and cannot resolve finer differences among genes.

Rather than the KEGG vs. gene variant conclusion, maybe a more interesting take-home message is how the communities varied with temperature and latitude. I was surprised the Abstract doesn't mention these patterns (it does mention temperature and distance differences), and the paper now has no graph of, say, richness vs. latitude. I believe the authors have some of the best data about these patterns, which would interest many ecologists, not just microbial oceanographers.

I hope the authors seriously consider the suggestions below about deleting some figures and text. It would help focus the paper on the most novel, interesting data. Some sections have too much speculation, as pointed out below in more detail.

One important technical detail: I don't think the authors say how they defined the taxa, as mentioned below. I suspect they used more than just 16S rRNA genes.

Specific comments

L4: Better: "the sunlit Atlantic Ocean microbiome" or "the sunlit microbiome in the Atlantic Ocean." Also, I think the authors should say "surface" or "euphotic" rather than "sunlit." The term is used only a couple times in the entire paper. Its use in the title sort of implies that the authors will discuss light effects, but they never do.

L36: I think the authors here mean "bacteria", not all microbes. If so, they should say bacteria or perhaps prokaryotes, to be clear. The "pelagic eukaryotic taxa" mentioned in the previous sentence also includes microbes.

L56: The phrase, "either highest diversity around 15 C or at highest temperature" isn't clear. Higher diversity at a temperature higher than 15 C?

L102: I'm not sure what the authors mean by the statement that excluding the unknown genes had little effect on the "overall structure of the AOM." What is meant by "structure" here? The authors could include the unknown genes for exploring how metagenomes differ along the transect, although they have to focus on the known genes for discussing how function varied.

L106: The authors state that bacterial production didn't vary consistently among the provinces without citing a figure or anything.

L113 and elsewhere: How were the taxa defined and delineated? By 16S genes or something else? I don't think this is discussed in the Methods section.

L117: I think the discussion about the "Effective Number" index and the related parts of Figure 2

should be deleted. The richness index has clearer patterns with temperature. The authors say that they used Effective Number index to avoid log scales, but I don't see what's so bad about a log scale. In any case, a log scale wasn't necessary for the richness index.

L126: The authors should give the correlation coefficient, not just the p-value.

L127: Based on an insignificant correlation, the authors have a lot of speculation. They say it indicates that "at intermediate temperatures of 15° to 20°C and seasonally fluctuating environmental and biotic conditions functional diversification of the microbial communities is enhanced. Such conditions have been shown to favor HGT and diversification and prevent microbes from eliminating functional genes, which may be discarded under more stable conditions to streamline genomes." Most of this doesn't make sense to me, and I don't see how they can say all that based on an insignificant correlation or even the significant relationships they see in their data.

L148: I don't think the authors need to define "WTRA" and "NATR" here, but they should say that they are oceanic provinces.

L153: The authors say that the NMDS plots show a "clear grouping" for the taxa and the nr genes and a "coarser grouping" for the KO profiles, but that's not clear to me. Is there any way to quantify the grouping, so the authors aren't relying on the reader seeing in the plots what they see? Average distance separating the groups in the NMDS plots?

L158: Because of what I should pointed out, I'm not convinced the following is correct: "The taxonomic and especially nr gene profiles reflect biogeographic provinces more accurately with less overlap than the KO profiles."

L161: It is telling that the authors didn't discuss the PERMANOVA results for the KO profiles. The r^2 values for these profiles were as high or sometimes higher than .

L184: I don't see how the authors can use their data to speculate as they do here, starting with "Diversification and adaptation of microbes to new ecological niches..." Just delete it.

L203: The authors should give the mean temperature for the eukaryotes so that readers can compare it with the one for prokaryotes. The authors should do a statistical test to see if the difference is significant.

L227: I don't see the need for this paragraph and Figure 5. It should be deleted and the space for the figure used for something else.

L257: I don't see the value of this paragraph—too much speculation. I think it should be deleted.

L278: It would be really interesting to see if the CAZyme genes are unique and do something different than say protein degradation genes.

L287: It was surprising and confusing to see cyanobacteria mentioned among the taxa with CAZyme genes. Only later do the authors imply that these genes are for internal carbohydrate processing, not like their role in heterotrophs. I think cyanobacteria shouldn't be mentioned until later, where there's space to explain how their CAZyme genes differ.

L289: The sentence starting with "CAZyme genes affiliated..." appears to be missing a verb.

L311: Here and elsewhere, the authors mention seasonality. I don't see how they can say anything about seasons when they don't have any data about how the communities change over time.

L360: Here and elsewhere: "million" shouldn't be capitalized.

Figure 1c: What is "CAZyme and Kegg"? Really needed?

Figure 1d: What is meant by "present in OM RGC v2"? Better to replace with something that can be immediately understood without having to look at the figure caption. For example, "novel sequence, function known" is clear.

Figure 2: As argued before, I think the authors should delete the Effective Number graphs. They don't add anything more (and only confuse the reader) than seen with the richness index. For panel j with the Permanova results, the authors give the r^2 numbers (or something like <0.1) even if insignificant. Now the readers have to guess at the meaning of the blank.

Figure 3: The size of the symbols and the thickness of the line should be reduced.

Figure 4: I don't understand what the shapes used to represent the data mean. They aren't explained in the caption. I urge the authors to use a more common box and whisker format, like that used in Fig 6C.

The authors should try to label which taxa are eukaryotes because the text compares them with the prokaryotes.

Figure 5: This should be deleted.

Figure 6A: What "quadratic equation" was used to fit the data? Why a quadratic? It seems arbitrary, especially so given there have been many attempts to describe how a property varies the temperature, starting with the Arrhenius relationship, which is not a quadratic.

It would be more straightforward and informative if the authors just give the correlation with temperature. I realize the relationship is probably not linear, but the correlation gives the reader the sense of the direction of the relationship.

Maybe better not to give the relationship with temperature at all in this figure. Put it elsewhere?

Figure 6C: The authors should use a more informative y-axis label, like "Relative abundance" and then explain what it means.

It took me some time to realize that the color in this figure matches (but not exactly) the colors given in panel b. I think the cluster numbers should be added to both panel b and c. To make room in panel b, the sample numbers (e.g. $n=51$) can be moved to the figure caption.

Finally, why does each graph in Fig 6C have both a box-and-whisker plot and a line plot? Why bother lumping data within a temperature range together? I think they authors should have only a line plot—temperature is a continuous variable.

Reviewer #2:

Remarks to the Author:

This study performed by Dlugosch and coauthors aimed to explore the functional biogeography of the surface microbiome of the Atlantic Ocean across a latitudinal gradient along a transect of 13,000 Km covering a nice temperature gradient (from 1 to 28°C) and covering nine contrasted biogeographical provinces based on clustering analyses of several variables. They conducted a series of fine analyses building first the Atlantic Ocean Microbiome (AOM) Gene Catalogue with a total of 7.75 Million nr-gene with 23% of novel genes of unknown functions. Also, they compared with the Tara Ocean gene catalogue displaying an overlapping of 59% reflecting a significant fraction of new genes uncovering by the AOM gene catalogue. Secondly, they explored general patterns in relationships with the temperature using different datasets based on taxonomy, KO and gene variants. They showed that richness of taxa, KOs and nr genes were significantly correlated to temperature with a maximum in subtropical regions around 15-20°C. They also displayed nice patterns of distance-decay relationships across the temperature and geographical distance. Until here, all patterns and data is well done and presented and although their findings are not extremely novel they present nicely the uncoupled between taxonomy vs functional diversity

based on KOs with extent functional redundancy. The relationship of the Bray Curtis dissimilarity (%) with the nr gene variants is larger is expected but again it is nice to see the increase with the temperature. Finally, they presented some results related to the mean temperature range of KO nr gene variants in abundant species and functional gene categories of the AOM (Fig. 4) but later were explained vaguely, specifically the findings associated with general KO categories because they used broad categories rather more resolute KEGG level.

In summary, I feel that the manuscript has solid results and interesting findings despite the novelty of this manuscript at its current version is moderate probably because of the way they presented some of their findings and some of their hypothesis such as in Line 243-347 "distinct distribution patterns of gene variants reflect different well-adapted populations of very closely related organisms and that the functional profile of ecotypes is not only modulated by the acquisition or loss of genes but also on sequence variation resulting in the selection for genes with a higher competitiveness in their respective environment" is not well supported by their current results. In this line, having MAGs reconstructed to further link the functional diversity of closely related MAGs with different distribution patterns (potential ecotypes) would be more informative.

Therefore it would be interesting to perform some extra analyses to extract more meaningful biological information. In that sense, I will appreciate some inputs on the following aspects.

1) Line 151-153. The authors mentioned that (NMDS) analysis based on Bray-Curtis dissimilarity showed a clear grouping of the communities both on the basis of taxonomy and nr genes, closely reflecting the biogeographic provinces. To me, the next analyses to present should be any enrichment analyses of which taxa and functions are enriched within those 9 biogeographical providences that emerged after the clustering analyses.

2) Lines 314-315. It would be much more interesting and informative to perform the co-assembly of the metagenomes within these 9 biogeographical provinces and explore the functional redundancy between closely related MAGs among different taxa. Doing so it would be possible to better infer the functional redundancy across taxa and the genetic microdiversity impact into large scale biogeographical patterns and functional processes in the Atlantic Ocean Microbiome.

Reviewer #3:

Remarks to the Author:

General Comments:

In the submitted article, "Significance of gene variants for the functional biogeography of the sunlit Atlantic Ocean's microbiome," authors Dlugosch et al examine the taxonomic and functional annotation of microbial metagenomes collected from 22 stations on two cruise transects across a latitudinal gradient in the Atlantic Ocean. The author's major claim is that gene variants, and not KEGG-orthologues, shape the functional biogeography of basin-scale marine microbiomes. They conclude that functional adaptation is evident on the level of gene variants, and thus their analysis demonstrates that niche adaptation by microbes is carried out by modification of functional genes.

I found many of the results presented in this article to be transformative for the field of marine microbial ecology. Their findings that (1) the diversity of functional genes peaks at 15-20C and largely mirrors patterns in taxonomic diversity (Figure 1), (2) the relative abundance of functional genes shows systematic and quantitative spatial variability associated with changing biogeographic regions (Figure 4), and (3) the relative abundance functional genes cluster by their spatial relationship with temperature (Figure 5) are particularly transformative. This work directly contradicts previous conclusions that the surface marine microbiome has relatively stable functional composition (i.e., Sunagawa et al 2015). Thus, this research has the potential to transform our understanding of the distribution of functional traits in the marine environment.

However, I also found that analysis of these results often lacked nuance and that the authors tended to describe their findings in ways that were not novel or convincing. I will provide three main examples and areas where the authors might improve:

First, in their abstract, the authors state, "gene variants, reflecting the fine-tuned adaptation of species ecotypes to regional biotic and environmental conditions, and not KOs, shape the functional biogeography of the Atlantic Ocean microbiome." In addition, the authors state, "The taxonomic and especially nr gene profiles reflect biogeographic provinces more accurately with less overlap than the KO profiles." However, these are methodological rather than substantive arguments. Because the taxonomic and KO profiles are based on database annotations, they necessarily represent a subset of sequences used for the nr gene variant profile. Thus, given the more limited dataset variability for the taxon and KO profiles, it is unsurprising that the nr gene variants data were more able to describe differences between samples. Moreover, although not all nr gene variants were annotated as KEGG orthologs, most are likely part of some orthologous group. Thus, I do not understand the argument that KOs do not "shape" the functional biogeography of the marine microbiome.

Second, a major conclusion of the authors is that microbial communities are well adapted to their environments and do so through the modification of functional genes. This conclusion, although undoubtedly true, is not particularly novel (e.g., see Larkin et al 2017). I think the authors would do well to place a greater emphasis on the novelty of the patterns observed here, such as the systematic and quantitative variability in functional genes across spatial gradients, which reveals significant differences in functional profiles across biogeographic regions and is strongly related to temperature.

Third, the authors state that "the affiliation to biogeographic provinces explained most of the variance within taxonomic, KO, and nr gene datasets... [suggesting] that holistic ecological features... better explain the large-scale structuring of oceanic microbial communities than single environmental variables." However, this conclusion is also not particularly novel. Longhurst provinces have long been used to describe the biogeography of the ocean based on multiple biogeochemical features. Moreover, biogeographic provinces have been regularly re-evaluated based on new biological data (e.g., Sonnewald et al 2020). Thus, multiple environmental factors are undoubtedly affecting the patterns observed here in multidimensional, interactive, and non-linear ways. Methods that may allow the authors to identify which environmental factors are most important for delineating functional trends, while also characterizing non-linear relationships, include general additive modeling (GAMs), neural networks, and/or random forest modeling. Such methods would allow for a more powerful assessment of the relationships between the environmental variables measured and either nr gene variant/taxon/KO diversity or composition, as compared to the results generated by perMANOVA and NMDS.

In terms of methodology, the techniques used to generate and analyze the dataset presented here were largely in line with the standards of the field. After initial quality control and contig assembly, representative non-redundant (nr) sequences identified by clustering at 95% identity using USEARCH. The taxonomy of the non-redundant sequences taxon was identified by Kaiju using the NCBI "nr" database. The functional annotation of non-redundant sequences was identified by GhostKOALA using the "KEGG" database and by DIAMOND using CAZy database. However, I worry that the rarefaction-based normalization technique used after the annotation steps may not have adequately accounted for differences in sequencing depth, which may in turn have affected some of the results presented in Figure 3. See below for detail.

Overall, I feel that this work has strong potential to be transformative. However, a more careful interpretation of the results, a greater emphasis on the novelty of the trends observed, and slightly more sophisticated analysis of the biogeographic and biogeochemical drivers of the functional profiles presented here would significantly improve this manuscript.

Specific Comments:

Line 33: Comma after "boundary conditions"

Line 35: constraints "on"

Lines 45-47: It is unclear to what the authors are referring when they state the "more refined trait

of a given gene orthologue" and "an environmentally well adapted gene or function." Gene content imbues microbes with specific traits, what does it mean for a trait to become more refined? Also, how would the authors define an environmentally "well adapted" function?

Line 64: "in the sunlit Atlantic Ocean, including a section in the Southern Ocean,"

Line 106-109: Heterotrophic prokaryotic production is described here (as well as in the methods), but no data, results, or analysis of this data is provided in either the main text or the supplements. In addition, there is no citation to support this statement.

Line 109-112: It is somewhat hard to distinguish the colors, but it looks as if the green-red stations (NADR-SATR) form one cluster and the purple-blue stations (FKLD-APLR) form a separate cluster. The authors should clarify what they mean by "stations of the corresponding provinces of the southern and northern hemisphere formed one cluster."

Line 153: The NMDS algorithm is designed to accurately represent a distance matrix. Unlike PCA or CCA, it does not automatically ensure the first NMDS axis lies along the direction of maximum variation. Consequently, rotating an NMDS solution should not change our interpretation of the result. Thus, if you rotate the KO NMDS clockwise 90 degrees, I believe the result would look quite similar to the other two NMDS plots.

Line 158-160: Is this surprising? We have more comprehensive taxonomic reference databases as compared to gene function databases for marine microbiomes. In addition, in Fig 1C the authors show that they have a greater % of reads with taxonomic classifications compared to functional annotations.

Line 164-167: The authors acknowledge the fact that their prescribed regions do a better job of predicting microbial structure compared to a single variable is an obvious conclusion. Thus, I am having trouble determining how this discussion adds significantly to our interpretation of the results.

Line 200-201/Figure 3: I am somewhat skeptical of these results. For example, *Prochlorococcus* abundance largely drops below detection in genomic samples taken below 10C and has also been shown to stop growing below 10C in laboratory cultures (T optimum for *Prochlorococcus* is ~25C, see Johnson et al 2006 and Flombaum et al 2013). Thus, it is surprising to see that the KO temperature range for *Prochlorococcus* is in the 7-10C range. In fact, unless I am misinterpreting the use of the terms "KO/nr gene variants, this result seems to contradict with the temperature range identified for *Prochlorococcus* (and *Synechococcus*) in Figure S7.

Thus, Figure 3 may highlight a potential statistical issue in the analysis. The authors chose to rarefy their dataset *after* classifying their metagenomic sequences into the nr, taxonomic, and KO clusters/annotations (rather than rarefying sequence count *before* clustering/annotation). Although this method is not fundamentally flawed, it does not account for the fact that differing sequencing depth between samples may skew the taxonomic composition observed. I worry that low sequencing depth at higher latitudes / lower temperatures, as well as an over-abundance of *Prochlorococcus* sequences in the reference database, may be over-inflating the percentage of sequences annotated as *Prochlorococcus* in the low temperature samples. One way to account for this would be to normalize coverage (or reads per kilobase) by taxon-specific single copy core gene coverage or to set a minimum coverage cutoff for the KO's analyzed here.

Line 390: "extent"

Lines 437-462: Supplementary materials list does not match supplementary materials provided

Figure 1c: Although the difference between "CAZymes & KEGG" vs "CAZymes" is described in the Methods, it is somewhat confusing without a corresponding description in the figure legend.

Figure 2: The font is quite small here. Ensure that font size conforms to journal standards.

Prof. Dr. Meinhard Simon
Institute for Chemistry and Biology of the Marine Environment
University of Oldenburg
D-26129 Oldenburg
Email: m.simon@icbm.de

Dear reviewers,

Thank you very much for your detailed and very constructive criticism and suggestions how to improve this study and clarify certain issues. As you will see in the revised manuscript and below we carried out further statistical evaluations to further strengthen our results on the distinct functional biogeography of microbial communities in the Atlantic and Southern Ocean and that gene variants of certain KOs in fact are the main influential factors of the biogeographic patterns observed. We replaced two of the old Figures by new ones showing these analyses.

Reviewer #1 (Remarks to the Author):

This paper explores an incredible data set of metagenomes from mostly bacteria along a 13,000 km transect from the South Atlantic (62 S) to the North Atlantic (47 N). Although they discuss briefly other data, such as chlorophyll and nutrients, they focus on temperature and distance to explore how bacterial communities varied along the transect. As stated in the last sentence of the Abstract, the authors' main conclusion is that "gene variants," not KEGG-orthologues, "shape the functional biogeography of the Atlantic microbiome."

I think the authors can come up with a better conclusion and a more novel "take-home" message. First, although Figure 3 clearly shows differences among communities defined by gene variants, Figure 2, the other main argument for these differences, is not as convincing, as argued below in more detail.

Second, even if the authors are correct, which I'm sure they are, I doubt many readers would be surprised to hear that differences between communities were more evident among gene variants than among KEGG-orthologues. KEGG is broad and cannot resolve finer differences among genes.

Rather than the KEGG vs. gene variant conclusion, maybe a more interesting take-home message is how the communities varied with temperature and latitude. I was surprised the Abstract doesn't mention these patterns (it does mention temperature and distance differences), and the paper now has no graph of, say, richness vs. latitude. I believe the authors have some of the best data about these patterns, which would interest many ecologists, not just microbial oceanographers.

We appreciate your valuable considerations. Based on our new analyses we modified the Abstract. We added a sentence on the relationship of richness and diversity with latitude and temperature. We did not want to take this as the take home message but put more emphasis on our new random forest modelling analyses and modified the last sentences with a more appropriate take home message.

I hope the authors seriously consider the suggestions below about deleting some figures and text. It would help focus the paper on the most novel, interesting data. Some sections have too much

speculation, as pointed out below in more detail.

One important technical detail: I don't think the authors say how they defined the taxa, as mentioned below. I suspect they used more than just 16S rRNA genes.

Thank you for pointing out this important detail. We now specify how we define the taxa in l. 78-84. As outlined our approach is based on the taxonomic binning of each gene to the closest gene in a reference genome. It may overestimate the total number of taxa as it distinguishes among closely related genomes which may be considered to belong to the same taxon based on other criteria.

Specific comments

L4: Better: "the sunlit Atlantic Ocean microbiome" or "the sunlit microbiome in the Atlantic Ocean." Also, I think the authors should say "surface" or "euphotic" rather than "sunlit." The term is used only a couple times in the entire paper. Its use in the title sort of implies that the authors will discuss light effects, but they never do.

We agree and changed the term to "near-surface Atlantic Ocean microbiome" throughout the manuscript.

L36: I think the authors here mean "bacteria", not all microbes. If so, they should say bacteria or perhaps prokaryotes, to be clear. The "pelagic eukaryotic taxa" mentioned in the previous sentence also includes microbes.

We changed the term to prokaryotes (now l. 37)

L56: The phrase, "either highest diversity around 15 C or at highest temperature" isn't clear. Higher diversity at a temperature higher than 15 C?

We specified this sentence as follows: However, different relationships have been reported, ranging from highest diversity at intermediate temperatures around 15°C to maxima between 25 and 30°C (l. 54-56).

L102: I'm not sure what the authors mean by the statement that excluding the unknown genes had little effect on the "overall structure of the AOM." What is meant by "structure" here? The authors could include the unknown genes for exploring how metagenomes differ along the transect, although they have to focus on the known genes for discussing how function varied.

Good point. We removed these sentences and the Supplementary Figures as they seem to be more confusing than helpful. In earlier versions of this manuscript submitted elsewhere we were criticized for either taking only a subset of classified genes or all of the available genes (which is mutually exclusive and we had to choose one of both options). Hence these data were added to show that clustering and overall Bray-Curtis distances of samples does not change substantially when excluding unclassified genes.

L106: The authors state that bacterial production didn't vary consistently among the provinces without citing a figure or anything.

We apologize for this missing data. We added data on bacterial production to Table S1 and Figure S3 and a citation in the methods text.

L113 and elsewhere: How were the taxa defined and delineated? By 16S genes or something else? I don't think this is discussed in the Methods section.

See above for how we delineated taxa (last point before Specific Comments).

L117: I think the discussion about the “Effective Number” index and the related parts of Figure 2 should be deleted. The richness index has clearer patterns with temperature.

The authors say that they used Effective Number index to avoid log scales, but I don’t see what’s so bad about a log scale. In any case, a log scale wasn’t necessary for the richness index.

We deleted the term Effective Number and now present richness and Shannon diversity. However, we normalized richness and diversity values to improve comparability between the two. Fig. 2a-f was modified accordingly (now l. 114 and below).

Just one remark to the log scale and Effective Number. Even though everybody presumably is aware of the log-scale it is not always intuitive that there is a strong difference between a Shannon Index of, e.g. 4.70 and 4.85 relative to a difference of 5.70 and 5.85. Using the Effective Number makes such a difference directly evident.

L126: The authors should give the correlation coefficient, not just the p-value.

We added the Pearson correlation coefficients. (now l. 130)

L127: Based on an insignificant correlation, the authors have a lot of speculation. They say it indicates that “at intermediate temperatures of 15° to 20°C and seasonally fluctuating environmental and biotic conditions functional diversification of the microbial communities is enhanced. Such conditions have been shown to favor HGT and diversification and prevent microbes from eliminating functional genes, which may be discarded under more stable conditions to streamline genomes.” Most of this doesn’t make sense to me, and I don’t see how they can say all that based on an insignificant correlation or even the significant relationships they see in their data.

Thank you for these considerations which made us thinking how to better explain and present what we intended to express. This insignificant Pearson correlation between richness and diversity and sequencing effort is just a support of our considerations that the high diversity and richness of the functional profiles at intermediate temperature favor microdiversification. Our main point for the discussion here is that KO and gene profiles exhibited different latitudinal patterns. Our discussion on the point that the highest richness and diversity at intermediate temperatures in temperate regions considers that extensive data document that these regions undergo large seasonal fluctuations in temperature, nutrient supply due to winter mixing and thus favour large fluctuations in plankton dynamics. These fluctuations are well documented by satellite-based data of chlorophyll, by data on nutrients, chlorophyll and other variables collected during many cruises by colleagues from the UK carried out between April and November in different years along latitudinal transects over the Atlantic (Atlantic Meridional Transect, AMT, Aiken et al. 2017, Robinson et al.2006). Further, analyses considering the interannual variation in sea surface temperature were also carried out by the Tara Ocean team (Ibarbalz et al., their Fig. 1A) and they did not find a correlation of this variable with the diversity of plankton groups. They did not examine gene diversity. From ecology and the concept of ecotones as the transition zones between adjacent ecosystems with usually a higher biodiversity diversity than within one ecosystem our considerations of maintaining a higher genetic functional diversity in the temperate regions with seasonally fluctuating environmental and biotic conditions is well supported, as we believe.

l. 148: I don’t think the authors need to define “WTRA” and “NATR” here, but they should say that they are oceanic provinces.

We are not exactly sure what you mean here. The two provinces are just given as examples that several functions were restricted to distinct lineages. Abbreviations of the province are defined in the legend of Fig. 1.

L153: The authors say that the NMDS plots show a “clear grouping” for the taxa and the nr genes and a “coarser grouping” for the KO profiles, but that’s not clear to me. Is there any way to quantify the grouping, so the authors aren’t relying on the reader seeing in the plots what they see? Average distance separating the groups in the NMDS plots?

Thank you for this valid critique. It prompted us to carry out further analyses to make this point clearer. We validated the clusters from all 3 datasets using the silhouette coefficient and comparisons of in-cluster vs. out-cluster distances. These results now clearly show that the taxonomic, KO and genes clusters differ, not only in the number of subclusters but also in the distances among the subclusters of each cluster (Fig. 2g-I, l. revision of text from l. 132-172). We added the details to the methods and Fig S5 (l. 423-425).

L158: Because of what I should pointed out, I’m not convinced the following is correct: “The taxonomic and especially nr gene profiles reflect biogeographic provinces more accurately with less overlap than the KO profiles.”

Thank you also for this valid critique. It prompted us to further analyses and we validated our point by another cluster-analysis using the silhouette coefficient (Fig. S5).

L161: It is telling that the authors didn’t discuss the PERMANOVA results for the KO profiles. The r^2 values for these profiles were as high or sometimes higher than.

Thank you also for this critique. Encouraged by a suggestion of Reviewer 3, we decided to use random forest models instead of the PERMANOVA to identify variables affecting the profiles of taxonomy, KOs and genes. These new analyses are presented in Fig. 2j-I and in the text from l. 136-172. The distinct impact of the different variables including provinces on the gene profile is discussed in l. 156-172. Using this modelling was added to the methods (l. 425-432).

L184: I don’t see how the authors can use their data to speculate as they do here, starting with “Diversification and adaptation of microbes to new ecological niches...” Just delete it.

Thank you for this critique. We realize that our data do not directly provide evidence for these considerations. However, we are convinced that it is important to include such considerations of how populations adapt to current or evolving environmental and biotic conditions in ecosystems and by which genetic means they may respond. Therefore we want to keep these arguments. Due the comprehensive revisions these sentences now appear at l. 236-248.

L203: The authors should give the mean temperature for the eukaryotes so that readers can compare it with the one for prokaryotes. The authors should do a statistical test to see if the difference is significant.

We agree with your comment. Due to the greatly different number of prokaryotic vs. eukaryotic taxa contained in this dataset, statistical analyses testing between these groups are not reliable (we tested it anyway using a Kruskal Wallis test and there was no statistical difference between the groups). To avoid speculation and confusion, we removed this statement.

L227: I don’t see the need for this paragraph and Figure 5. It should be deleted and the space for the figure used for something else.

We agree and revised the entire paragraph comprehensively and moved Fig. 5 to the supplement (new Fig. S8).

L257: I don’t see the value of this paragraph—too much speculation. I think it should be deleted.

We do understand your arguments about these speculations. However, we believe that it is important to include certain valuable speculations on some important results in a publication which

may inspire at least some readers to further analyses. As you stated earlier our study is based on a very reliable and comprehensive data set which should be a good basis for such studies and further analyses. Therefore we want to keep these speculations.

L278: It would be really interesting to see if the CAZyme genes are unique and do something different than say protein degradation genes.

We completely agree with your argument and would have loved to go further with our analyses. However, the exact classification of CAZymes and their substrates is still an issue and mostly relies on the surrounding operon (Krüger et al. 2019, ISME J; Francis et al 2021, ISME J). Other than that they are mostly characterized by structural similarity. Our results show that there seems to be systematic change in their abundance, probably to adjust for different and unfortunately unidentified substrates in the environment. To put the CAZyme chapter into a better context we moved it to an earlier part of the manuscript (now l. 173-219).

L287: It was surprising and confusing to see cyanobacteria mentioned among the taxa with CAZyme genes. Only later do the authors imply that these genes are for internal carbohydrate processing, not like their role in heterotrophs. I think cyanobacteria shouldn't be mentioned until later, where there's space to explain how their CAZyme genes differ.

We revised the entire chapter on CAZyme families and moved it to an earlier section of the manuscript (now l. 173-219). We mention now the link of CAZymes of Cyanobacteria to photosynthesis at the first mentioning of Cyanobacteria. Further, we made a new analysis of the occurrence of genes encoding CAZymes in the three clusters of the functional profiles to show relative enrichments of certain CAZyme families (new Fig. 3b).

L289: The sentence starting with "CAZyme genes affiliated..." appears to be missing a verb.

Thank you. We added "were" to complete the sentence (now l. 208-209).

L311: Here and elsewhere, the authors mention seasonality. I don't see how they can say anything about seasons when they don't have any data about how the communities change over time.

We do understand your point. However, as also pointed out above, there is good evidence of seasonal variations of environmental and biotic variables and plankton dynamics in the temperate regions of the Atlantic on which we can rely (AMT cruises, Aiken et al. 2017, Robinson et al. 2006, Longhurst 2006, Reygondeau et al 2013). Therefore we feel confident to be able to speculate on possible reasons for the observed patterns to inspire future research on the impact of seasonal dynamics on the stability and functionality of microbial communities in the Atlantic Ocean and beyond.

L360: Here and elsewhere: "million" shouldn't be capitalized.

These changes are made (l. 370 and elsewhere).

Figure 1c: What is "CAZyme and Kegg"? Really needed?

We now pooled KEGG and CAZyme to simplify the legend.

Figure 1d: What is meant by "present in OM RGC v2"? Better to replace with something that can be immediately understood without having to look at the figure caption. For example, "novel sequence, function known" is clear.

We simplified the legend and moved the reference to the MG RGC v2 to the Figure capture.

Figure 2: As argued before, I think the authors should delete the Effective Number graphs. They don't add anything more (and only confuse the reader) than seen with the richness index. For panel j with the Permanova results, the authors give the r^2 numbers (or something like <0.1) even if insignificant. Now the readers have to guess at the meaning of the blank.

According to the new analyses including Shannon diversity we modified Fig. 2a-f and removed any reference to Effective Number. Now normalized richness/diversity over latitudinal and temperature gradient is shown. As mentioned before, PERMANOVA is substituted by random forest models to determine influence of environmental parameters to purity of functional clusters.

Figure 3: The size of the symbols and the thickness of the line should be reduced.

Changed as requested (now Fig. 5).

Figure 4: I don't understand what the shapes used to represent the data mean. They aren't explained in the caption. I urge the authors to use a more common box and whisker format, like that used in Fig 6C. The authors should try to label which taxa are eukaryotes because the text compares them with the prokaryotes.

Thank you for your critique. We redraw the entire Figure (now Fig. 6) and replaced the Violin-plot by traditional Box-plots. A colour code was added for a clear distinction of the different taxonomic groups of prokaryotes and eukaryotes.

Figure 5: This should be deleted.

As pointed out above we moved this Figure to the Supplement (Fig. S8).

Figure 6A: What "quadratic equation" was used to fit the data? Why a quadratic? It seems arbitrary, especially so given there have been many attempts to describe how a property varies the temperature, starting with the Arrhenius relationship, which is not a quadratic.

Although the Arrhenius relationship describes enzymatic properties in relationship to temperature, it does not necessarily fit for abundances of certain genes. To avoid overfitting, we used unimodal fits as they were the simplest way to describe all observed CAZyme abundance patterns. Cluster one and three (increasing or decreasing with T) are near-linear while cluster two is hump shaped. For methodological consistency we chose to use the same approach for all clusters.

It would be more straightforward and informative if the authors just give the correlation with temperature. I realize the relationship is probably not linear, but the correlation gives the reader the sense of the direction of the relationship. Maybe better not to give the relationship with temperature at all in this figure. Put it elsewhere?

We think that the direction of the relationship is clear in Fig. 4. Pearson/Spearman correlation would not be able to show the hump-shaped correlation between CAZyme cluster 2 and ambient temperature.

Figure 6C: The authors should use a more informative y-axis label, like "Relative abundance" and then explain what it means. It took me some time to realize that the color in this figure matches (but not exactly) the colors given in panel b. I think the cluster numbers should be added to both panel b and c. To make room in panel b, the sample numbers (e.g. n=51) can be moved to the figure caption. Finally, why does each graph in Fig 6C have both a box-and-whisker plot and a line plot? Why bother lumping data within a temperature range together? I think they authors should have only a line plot—temperature is a continuous variable.

Thank you for your suggestions to improve the readability of this Figure. We modified panel C and changed the axis label to "normalised CAZyme abundance" and added the number of CAZyme

families to the Figure capture. The colours of panels B and C did not match 100% because the background colour of the boxplots was slightly transparent. We keep using the transparency by presenting all data points to indicate higher density of data in the new, non-boxplot version.

Reviewer #2 (Remarks to the Author):

This study performed by Dlugosch and coauthors aimed to explore the functional biogeography of the surface microbiome of the Atlantic Ocean across a latitudinal gradient along a transect of 13,000 Km covering a nice temperature gradient (from 1 to 28°C) and covering nine contrasted biogeographical provinces based on clustering analyses of several variables. They conducted a series of fine analyses building first the Atlantic Ocean Microbiome (AOM) Gene Catalogue with a total of 7.75 Million nr-gene with 23% of novel genes of unknown functions. Also, they compared with the Tara Ocean gene catalogue displaying an overlapping of 59% reflecting a significant fraction of new genes uncovered by the AOM gene catalogue. Secondly, they explored general patterns in relationships with the temperature using different datasets based on taxonomy, KO and gene variants. They showed that richness of taxa, KOs and nr genes were significantly correlated to temperature with a maximum in subtropical regions around 15-20°C. They also displayed nice patterns of distance-decay relationships across the temperature and geographical distance. Until here, all patterns and data is well done and presented and although their findings are not extremely novel they present nicely the uncoupled between taxonomy vs functional diversity based on KOs with extent functional redundancy. The relationship of the Bray Curtis dissimilarity (%) with the nr gene variants is larger is expected but again it is nice to see the increase with the temperature. Finally, they presented some results related to the mean temperature range of KO nr gene variants in abundant species and functional gene categories of the AOM (Fig. 4) but later were explained vaguely, specifically the findings associated with general KO categories because they used broad categories rather more resolutive KEGG level.

In summary, I feel that the manuscript has solid results and interesting findings despite the novelty of this manuscript at its current version is moderate probably because of the way they presented some of their findings and some of their hypothesis such as in Line 243-347 “distinct distribution patterns of gene variants reflect different well-adapted populations of very closely related organisms and that the functional profile of ecotypes is not only modulated by the acquisition or loss of genes but also on sequence variation resulting in the selection for genes with a higher competitiveness in their respective environment” is not well supported by their current results. In this line, having MAGs reconstructed to further link the functional diversity of closely related MAGs with different distribution patterns (potential ecotypes) would be more informative.

Therefore it would be interesting to perform some extra analyses to extract more meaningful biological information. In that sense, I will appreciate some inputs on the following aspects.

1) Line 151-153. The authors mentioned that (NMDS) analysis based on Bray-Curtis dissimilarity showed a clear grouping of the communities both on the basis of taxonomy and nr genes, closely reflecting the biogeographic provinces. To me, the next analyses to present should be any enrichment analyses of which taxa and functions are enriched within those 9 biogeographical provinces that emerged after the clustering analyses.

Thank you for your supporting assessment of our study and the suggestions to further improve the manuscript.

Inspired by your suggestions we carried out a refined cluster analysis to better delineate the taxonomic, KO and nr gene clustering and present this analysis in Fig. 1 g-i. It clearly shows that the gene cluster is distinct and more detailed than the others. We further extended the analysis to assess the differential abundance of genes in the functional clusters obtained from the KO dataset and included also the CAZymes encoding genes. A selection of functions and their enrichment in those

clusters is displayed in new Fig. 3 and shows that distinct functional categories are differently distributed between pairs of clusters. We modified the text accordingly (l. 148-186).

2) Lines 314-315. It would be much more interesting and informative to perform the co-assembly of the metagenomes within these 9 biogeographical provinces and explore the functional redundancy between closely related MAGs among different taxa. Doing so it would be possible to better infer the functional redundancy across taxa and the genetic microdiversity impact into large scale biogeographical patterns and functional processes in the Atlantic Ocean Microbiome.

Thank you for this suggestion. In fact, we had looked at MAGs earlier when we analysed the data but realised that we presumably had no representative distribution of MAGs among phylogenetic groups and provinces for a reliable analysis. With your suggestions we came back to these analyses and used metabat2, checkM and gtdb-tk to bin, quality-check and classify MAGs from our metagenomes but were not able to perform the suggested analysis, due to lack of closely related and widespread MAGs of decent quality. Only ~90 MAGs showed >80% completeness with less than 5% contamination and only very few of them belonged to ubiquitously abundant taxa. We are not confident that any analysis of those MAGs in this way would be scientifically sound and therefore did not include any such analyses into our study. Based on a larger data set, including also MAGs from other oceanic data sets, we will analyse these MAGs and publish these data as part of a different study.

Reviewer #3 (Remarks to the Author):

General Comments:

In the submitted article, “Significance of gene variants for the functional biogeography of the sunlit Atlantic Ocean’s microbiome,” authors Dlugosch et al examine the taxonomic and functional annotation of microbial metagenomes collected from 22 stations on two cruise transects across a latitudinal gradient in the Atlantic Ocean. The author’s major claim is that gene variants, and not KEGG-orthologues, shape the functional biogeography of basin-scale marine microbiomes. They conclude that functional adaptation is evident on the level of gene variants, and thus their analysis demonstrates that niche adaptation by microbes is carried out by modification of functional genes.

I found many of the results presented in this article to be transformative for the field of marine microbial ecology. Their findings that (1) the diversity of functional genes peaks at 15-20C and largely mirrors patterns in taxonomic diversity (Figure 1), (2) the relative abundance of functional genes shows systematic and quantitative spatial variability associated with changing biogeographic regions (Figure 4), and (3) the relative abundance functional genes cluster by their spatial relationship with temperature (Figure 5) are particularly transformative. This work directly contradicts previous conclusions that the surface marine microbiome has relatively stable functional composition (i.e., Sunagawa et al 2015). Thus, this research has the potential to transform our understanding of the distribution of functional traits in the marine environment.

Thank you for this generally positive but also critical review which prompted us to carry out further analyses and to rewrite certain parts of the manuscripts. Our responses to your specific points are written below these points.

However, I also found that analysis of these results often lacked nuance and that the authors tended to describe their findings in ways that were not novel or convincing. I will provide three main examples and areas where the authors might improve:

First, in their abstract, the authors state, “gene variants, reflecting the fine-tuned adaptation of species ecotypes to regional biotic and environmental conditions, and not KOs, shape the functional biogeography of the Atlantic Ocean microbiome.” In addition, the authors state, “The taxonomic and especially nr gene profiles reflect biogeographic provinces more accurately with less overlap than the KO profiles.” However, these are methodological rather than substantive arguments. Because the taxonomic and KO profiles are based on database annotations, they necessarily represent a subset of sequences used for the nr gene variant profile. Thus, given the more limited dataset variability for the taxon and KO profiles, it is unsurprising that the nr gene variants data were more able to describe differences between samples. Moreover, although not all nr gene variants were annotated as KEGG orthologs, most are likely part of some orthologous group. Thus, I do not understand the argument that KOs do not “shape” the functional biogeography of the marine microbiome.

We understand your statement that also the taxon and KO profiles shape the AOM along the transect. However, as pointed out in the text the nr gene profile (now termed only gene profile) provides a much higher resolution than the taxon and KO profiles which we now substantiate with a refined cluster analysis of the NMDS analysis (Fig. 2g-i). We do acknowledge that the two former profiles shape the AOM but we are not aware of any study providing such a refined analysis of gene profiles and how they reflect biogeographic patterns in microbial oceanography and presumably beyond.

Second, a major conclusion of the authors is that microbial communities are well adapted to their

environments and do so through the modification of functional genes. This conclusion, although undoubtedly true, is not particularly novel (e.g., see Larkin et al 2017). I think the authors would do well to place a greater emphasis on the novelty of the patterns observed here, such as the systematic and quantitative variability in functional genes across spatial gradients, which reveals significant differences in functional profiles across biogeographic regions and is strongly related to temperature.

We agree that that the concept of gene adaptation of ecological niches is not new. Especially the publication by Larkin et al. 2017 provides a comprehensive framework based on a modelling approach and data from pure culture work. To the best of our knowledge empirical data based on metagenomics analyses on such a high resolution level as we present are not available but urgently needed. We think our analyses provide a valuable proof of these concepts and additional data that can potentially explain observed richness/diversity patterns of the AOM.

In the revised version we tried to emphasize the novelty of our analyses at various places, also supported by the additional analyses we carried out and believe that these novelties are clearly described.

Third, the authors state that “the affiliation to biogeographic provinces explained most of the variance within taxonomic, KO, and nr gene datasets... [suggesting] that holistic ecological features... better explain the large-scale structuring of oceanic microbial communities than single environmental variables.” However, this conclusion is also not particularly novel. Longhurst provinces have long been used to describe the biogeography of the ocean based on multiple biogeochemical features. Moreover, biogeographic provinces have been regularly re-evaluated based on new biological data (e.g., Sonnewald et al 2020). Thus, multiple environmental factors are undoubtedly affecting the patterns observed here in multidimensional, interactive, and non-linear ways. Methods that may allow the authors to identify which environmental factors are most important for delineating functional trends, while also characterizing non-linear relationships, include general additive modeling (GAMs), neural networks, and/or random forest modeling. Such methods would allow for a more powerful assessment of the relationships between the environmental variables measured and either nr gene variant/taxon/KO diversity or composition, as compared to the results generated by perMANOVA and NMDS.

Thank you for these valuable considerations which also prompted us to carry out further analyses. One valuable aspect of our study is to include prokaryotic genomic functional features into the analysis of oceanographic biogeographic features and we were able to show that they match the existing biogeographic concepts. This is particularly evident by the random forest modelling analysis we carried out. Among others, these new results show that the prokaryotic genetically encoded functions are more influenced by biotic factors than temperature or inorganic nutrients, features hard to grasp by classical oceanographic and plankton-focused analyses. In contrast, cluster purity of the taxonomic dataset was affected predominantly by temperature (Fig. 2j-l, revised I. 132-172).

In terms of methodology, the techniques used to generate and analyze the dataset presented here were largely in line with the standards of the field. After initial quality control and contig assembly, representative non-redundant (nr) sequences identified by clustering at 95% identity using USEARCH. The taxonomy of the non-redundant sequences taxon was identified by Kaiju using the NCBI “nr” database. The functional annotation of non-redundant sequences was identified by GhostKOALA using the “KEGG” database and by DIAMOND using CAZy database. However, I worry that the rarefaction-based normalization technique used after the annotation steps may not have adequately accounted for differences in sequencing depth, which may in turn have affected some of the results presented in Figure 3. See below for detail.

Rarefaction: The datasets we used for all analysis besides the calculation of alpha diversity were not rarefied. We did this specifically for this analysis to have a comparable sample size for these calculations as vastly different sequencing depth might result in higher richness in unrarefied data.

Overall, I feel that this work has strong potential to be transformative. However, a more careful interpretation of the results, a greater emphasis on the novelty of the trends observed, and slightly more sophisticated analysis of the biogeographic and biogeochemical drivers of the functional profiles presented here would significantly improve this manuscript.

For the revision we carried out new analyses regarding the clustering of the taxonomic, KO and gene profiles in the biogeographic provinces including a random forest modelling as shown in Fig. 2j-l and revised the text accordingly. We believe that this revision considered most of your critique. For more specific answers to your critique see the detailed answer below.

Specific Comments:

Line 33: Comma after “boundary conditions”

Comma added (l. 34).

Line 35: constraints “on”

corrected (l. 36)

Lines 45-47: It is unclear to what the authors are referring when they state the “more refined trait of a given gene orthologue” and “an environmentally well adapted gene or function.” Gene content imbues microbes with specific traits, what does it mean for a trait to become more refined? Also, how would the authors define an environmentally “well adapted” function?

We replaced the term refined trait by specific trait (l. 44) and think that in the following specification, “kinetic features and/or temperature range and optima” we describe what is meant by the specific features. We believe that these specific descriptions define “an environmentally well adapted function.”

Line 64: “in the sunlit Atlantic Ocean, including a section in the Southern Ocean,”

First, we replaced sunlit by near-surface in the entire manuscript and modified the sentence to “In order to address these questions we investigated the taxonomic and functional diversity of microbial communities in the near-surface Atlantic Ocean, including a section in the Southern Ocean, along a 13,000 km transect between 62°S and 47°N”. (l. 63-65)

Line 106-109: Heterotrophic prokaryotic production is described here (as well as in the methods), but no data, results, or analysis of this data is provided in either the main text or the supplements. In addition, there is no citation to support this statement.

We renamed this term to bacterial biomass production, as the great majority of prokaryotes were Bacteria. And we added data to Table S1 and Figure S4 and to the Methods including a reference (l. 345-346).

Line 109-112: It is somewhat hard to distinguish the colors, but it looks as if the green-red stations (NADR-SATR) form one cluster and the purple-blue stations (FKLD-APLR) form a separate cluster. The authors should clarify what they mean by “stations of the corresponding provinces of the southern and northern hemisphere formed one cluster.”

In the revision we deleted this sentence but still show the clustering in the Supplement, Fig. S3. The revised cluster analysis now provides clear-cut patterns in the taxonomic, KO and gene profiles. Regarding the colors of the provinces we thought that they allow distinguishing among them, also as no other reviewer complaint about them.

Line 153: The NMDS algorithm is designed to accurately represent a distance matrix. Unlike PCA or CCA, it does not automatically ensure the first NMDS axis lies along the direction of maximum variation. Consequently, rotating an NMDS solution should not change our interpretation of the result. Thus, if you rotate the KO NMDS clockwise 90 degrees, I believe the result would look quite similar to the other two NMDS plots.

Thank you for this important critical comment. We addressed this issue with further analyses of clusters within the different datasets and their validation. We updated Fig. 2 and added Fig. S5 to show that the described trends are robust and not only based on visual observations.

Line 158-160: Is this surprising? We have more comprehensive taxonomic reference databases as compared to gene function databases for marine microbiomes. In addition, in Fig 1C the authors show that they have a greater % of reads with taxonomic classifications compared to functional annotations.

For the reasons mentioned in this comment it is not particularly surprising, but to the best of our knowledge this has not been shown before and we believe it is important to emphasize. Although it is true that a higher percentage of sequences in the AOM-RGC is classified taxonomically, the subset of data used for the analysis contains only sequences that are taxonomically and functionally classified.

Line 164-167: The authors acknowledge the fact that their prescribed regions do a better job of predicting microbial structure compared to a single variable is an obvious conclusion. Thus, I am having trouble determining how this discussion adds significantly to our interpretation of the results. We think that this point of discussion is important in particular because our study shows in a very comprehensive and, as we believe, empirically well-supported way how well the biogeography of oceanic microbial communities fit into the province concept introduced by Longhurst.

Line 200-201/Figure 3: I am somewhat skeptical of these results. For example, Prochlorococcus abundance largely drops below detection in genomic samples taken below 10C and has also been shown to stop growing below 10C in laboratory cultures (T optimum for Prochlorococcus is ~25C, see Johnson et al 2006 and Flombaum et al 2013). Thus, it is surprising to see that the KO temperature range for Prochlorococcus is in the 7-10C range. In fact, unless I am misinterpreting the use of the terms "KO/nr gene variants, this result seems to contradict with the temperature range identified for Prochlorococcus (and Synechococcus) in Figure S7.

Thus, Figure 3 may highlight a potential statistical issue in the analysis. The authors chose to rarefy their dataset *after* classifying their metagenomic sequences into the nr, taxonomic, and KO clusters/annotations (rather than rarefying sequence count *before* clustering/annotation). Although this method is not fundamentally flawed, it does not account for the fact that differing sequencing depth between samples may skew the taxonomic composition observed. I worry that low sequencing depth at higher latitudes / lower temperatures, as well as an over-abundance of Prochlorococcus sequences in the reference database, may be over-inflating the percentage of sequences annotated as Prochlorococcus in the low temperature samples. One way to account for this would be to normalize coverage (or reads per kilobase) by taxon-specific single copy core gene coverage or to set a minimum coverage cutoff for the KO's analyzed here.

Thank you for your detailed description of your skepticism on this issue. It told us that the data are not presented in a clear-cut way showing what we really meant. But we think that you wanted to refer to the old Fig. 4, not 3. There seems to be a misunderstanding in the interpretation of what is shown in this Figure and Fig. S7 (now S9): The overall temperature range we describe, e.g. for *Prochlorococcus marinus*/sp. is from ca. 14 to 28°C and 19 to 28, respectively, as shown in Figure S7 (now S9) equivalent to temperature ranges of 14 and 9 °C, respectively. These ranges and the absolute temperatures are in line with data from previous publications. We revised the caption and hope that it now becomes clear what the Figure shows.

The datasets we used for all analysis besides the calculation of alpha diversity were not rarefied. We did this specifically for this analysis to have a comparable sample size for these calculations as vastly different sequencing depth might result in higher richness in unrarefied data.

Samples were normalized by using a conversion to counts per million (analogous to transcripts per million used in transcriptomics) to account for the differences in sequencing depth. The presence of *Prochlorococcus* genes outside its known and well documented latitudinal range is probably due to misclassification of genes of *Prochlorococcus*. A recent benchmark study (Ye et al 2019, Cell; Benchmarking Metagenomics Tools for taxonomic classification) demonstrated that Kaiju performs about as well as other available tools with >90% accuracy on species level. The extremely low abundance of these misclassified genes however is unlikely to change our results in a meaningful way and we are confident that we observe real trends in the data despite methodologically unpreventable biases.

Line 390: “extent”

Corrected (l. 401).

Lines 437-462: Supplementary materials list does not match supplementary materials provided

Thank you for this careful check. We hope to have the correct list of Supplementary Tables and Figures in the revised version.

Figure 1c: Although the difference between “CAZymes & KEGG” vs “CAZymes” is described in the Methods, it is somewhat confusing without a corresponding description in the figure legend.

We modified this Figures considering your point.

Figure 2: The font is quite small here. Ensure that font size conforms to journal standards.

Corrected.

Reviewers' Comments:

Reviewer #1:

Remarks to the Author:

The authors followed many of the reviewer comments and suggestions, I believe, but there is still work to do.

It was difficult to evaluate the last third or so of the paper because there is a mismatch between the figures and the text, as pointed out below (L208, L232, and L256). It's as if a figure was deleted or something added but the figure numbers in the text weren't changed correctly.

The authors came up with a different "take home" message, but I think they could do better. Their observation about "fine-tuned genetic adaptation" is not surprising. More surprising is their observation given in the second to last sentence of the Abstract, about biotic factors vs. temperature and biogeographic provinces. As pointed out below, the authors could do a better job of highlighting that observation and discussing its implications.

The authors seem to have retained a lot of speculation. I urge them to delete as much as they can, so that readers can concentrate on the important stuff. Some suggestions are given below. They especially should delete speculation about a statistically insignificant relationship (L277).

The writing is difficult to follow at times. Sentences with "show" (and relatives) and "exhibited" would be much better if written without them. The authors introduce a lot of abbreviations, which can make it difficult for readers to follow the text. I point out below some cases where abbreviations aren't necessary.

Specific comments

L27: "and gene profiles" doesn't make sense here, after mentioning "abundance of genes involved in nutrient and energy acquisition."

L28: I don't remember seeing in the main text a detailed discussion of this very important conclusion about "biotic factors" setting the patterns in gene functions while temperature and biogeographic provinces set taxonomy. Or if it's there, it was obscured by other stuff. The authors should make sure future readers don't miss this.

L44: "KOs" should be defined—better, it should be replaced with something that could be understood by someone not familiar with KOs.

L129: The sentence beginning with "Richness and diversity were independent..." isn't clear. It's not clear what is being correlated.

L139: The variation in taxonomy and genes explained by the measured properties is very high, among the highest I remember seeing. Is that true, that it's very high? Maybe worthwhile emphasizing more. Why were the authors able to see such a high percentage whereas others were not as successful?

L167: The sentence beginning here ("Our findings further imply that...") is pure speculation without any connection to what the authors actually measured.

L182: It doesn't make sense that nitrification has such a big impact on defining these clusters, because nitrifiers are much less abundant than heterotrophs and cyanobacteria. Perhaps the authors can find other data indicating nitrifiers are more abundant than would be expected.

L189: There is no need to use abbreviations here for the CAZymes.

L208: The Figure 4 discussed here is not the Figure 4 given in the paper. The discussion is about how CAZyme clusters varied with temperature, but the actual Figure 4 shows how dissimilarities in taxonomy, KOs, and genes varied with distance and temperature.

L210: There is no need to introduce the abbreviation "cC."

L231: No need for "TD."

L232: The text should cite the current Figure 4, not Figure 5.

L242: The text following the sentence that starts here ("Recent findings of high gene variant...") is pure speculation not related to the authors' data. It should be deleted.

L256: The text should cite the current Figure 5, not Figure 6.

L277: The authors say that "different KOs vary in their temperature ranges and means" but then in the next sentence, they admit the differences are not significant. They still say the trends are "obvious and worth exploring." Maybe. The only thing "obvious" is the lack of any significant difference. That should be the conclusion, not that the KOs varied with temperature.

L279: Given the lack of a significant difference, the speculation about how different genes vary with temperature is not warranted and should be deleted.

L285: This speculation should be deleted.

Figure 2: The caption needs to explain how the lines were drawn and the meaning of the shading around each line. I think the shading should be removed—it only complicates trying to understand the graph.

Figure 3: For the x-axis label, "difference" seems more appropriate than "change" because what is plotted is the difference between two clusters, not a change over time or space. Also, how can there be two, e.g. "4-fold", for each I presume one is negative, but not sure how that is even possible.

Figure 6: The caption for panel a needs to explain the r^2 values. I don't see the need for panel b—it can be deleted or moved to supplemental materials. The tick labels for panels a and c should be rotated 90 degrees so readers don't have to rotate their head to read them. The trend lines in panel c are too thick.

The caption for panel c needs to explain why there are three patterns to how CAZyme abundance varies with temperature. These panels are really interesting.

Reviewer #2:

Remarks to the Author:

The authors have done a nice effort integrating most of the requests and questions raised by the different reviewers and the manuscript has now a much better structure and flow. They have answered properly to all my concerns, deriving a new figure (Fig3) and I found their current version well improved and suitable for publication.

Reviewer #3:

Remarks to the Author:

In the submitted article, "Significance of gene variants for the functional biogeography of the sunlit Atlantic Ocean's microbiome," authors Dlugosch et al examine the environmental structuring of microbial taxonomic and functional biogeography across the Atlantic Ocean basin. Their results are transformative for the fields of biological oceanography and microbial ecology in that they convincingly demonstrate that, at high genomic resolution, functional genes show systematic and quantitative spatial patterns with both biogeographic province and temperature.

Previously, my main criticism was that I felt the authors' interpretation of their results often lacked nuance or their results were not presented in a novel way. The authors have made a significant effort to refine their manuscript. As a result, the authors now present a more sophisticated analysis that is better contextualized in regards to the broader literature.

Moreover, I am very excited to see the authors' random forest analysis. Their conclusion that temperature better predicted taxonomic composition but that POC and chlorophyll better predicted functional profiles has very interesting implications for both the spatial and temporal structuring of microbial community composition and function.

Thus, the authors have addressed all my previous concerns. I have a few minor comments regarding grammar and clarity. Once these are addressed I have no further comments.

Minor comments:

Line 26-28: This sentence is a bit confusing. Perhaps split into two sentences, i.e., "A cluster analysis yielded three clusters of KOs but five clusters of genes differing in the abundance of genes involved in nutrient and energy acquisition. Gene profiles showed much higher distance-decay rates than KO and taxonomic profiles."

Line 29: Add comma- "observed patterns in the functional profiles, whereas temperature..."

Line 37-38: Move comma- "due to their high abundance and enormous taxonomic and functional diversity, attempts..."

Line 46: What is your exact definition of a "gene variant"? Ostensibly it is a gene associated with a specific function, but is it also associated with a specific taxon? And if so, what taxonomic "level" (i.e., 100% nucleotide similarity for a specific gene? 97% similarity?). Please clarify this term.

Line 105: Figure S2 has some typos and figure panel labelling errors in the caption

Line 137-140: Typo- "we fit random forest (rf) models"

Figure 2 caption: Typo- "determined by random forest models"

Although the captions for figures 4, 5, and 6 are correct, the order of the submitted images for these figures is shuffled

Reviewer #1 (Remarks to the Author):

The authors followed many of the reviewer comments and suggestions, I believe, but there is still work to do.

Thank you for this generally favourable remark acknowledging our successful efforts to improve the manuscript and for your careful reading of this revised version.

It was difficult to evaluate the last third or so of the paper because there is a mismatch between the figures and the text, as pointed out below (L208, L232, and L256). It's as if a figure was deleted or something added but the figure numbers in the text weren't changed correctly.

We apologize for these inconsistencies which are now fixed as outlined below.

The authors came up with a different "take home" message, but I think they could do better. Their observation about "fine-tuned genetic adaption" is not surprising. More surprising is their observation given in the second to last sentence of the Abstract, about biotic factors vs. temperature and biogeographic provinces. As pointed out below, the authors could do a better job of highlighting that observation and discussing its implications.

We think that the conclusions and take home message is rather well reflecting our main and novel findings. We think that the effect of biotic variables explaining the functional biogeographic patterns is definitely important but we also think that our analyses on the fine-tuned genetic adaptations are as important because, even though this might have assumed implicitly in many other studies, but has never been shown in such a way. We revised the Conclusion such that the biotic interactions are more highlighted (l. 321-324).

The authors seem to have retained a lot of speculation. I urge them to delete as much as they can, so that readers can concentrate on the important stuff. Some suggestions are given below. They especially should delete speculation about a statistically insignificant relationship (L277).

We can understand that our discussion regarding certain observations seems speculative on the one hand. On the other hand, such speculations need to be done and stimulate future work and thus are important triggers. We think that we are not too bold in our speculations and most of them are based on clear-cut and sound analyses. Two cases of perceived speculations are specified in the detailed comments below and supported by cited literature and refer to previously published studies used for the interpretation of our data. With regard to statements we make in the discussion, we believe that we do not overstate the significance (or the lack thereof) in the text and discuss the given results transparently enough to avoid false conclusions by the reader. We specifically avoid claiming significance where there is none while still discussing our results in the context of recent literature. To a certain degree speculating about the mechanisms on so far unobserved or unreported patterns is necessary to make sense of the results and give starting points for ecological studies in the future.

The writing is difficult to follow at times. Sentences with "show" (and relatives) and "exhibited" would be much better if written without them. The authors introduce a lot of abbreviations, which can make it difficult for readers to follow the text. I point out below some cases where abbreviations aren't necessary.

Our style is to write like “we show” etc. to make a distinction between our observations and well established facts. We would like to stick to this style which is also used by other authors.

We removed abbreviations for ‘temperature difference’, ‘CAZyme clusters’ and ‘random forest’ from the text. Some abbreviations are well introduced by other authors and hence were kept.

Specific comments

L27: “and gene profiles” doesn’t make sense here, after mentioning “abundance of genes involved in nutrient and energy acquisition.”

Sentence modified (l. 26-28).

L28: I don’t remember seeing in the main text a detailed discussion of this very importance conclusion about “biotic factors” setting the patterns in gene functions while temperature and biogeographic provinces set taxonomy. Or if it’s there, it was obscured by other stuff. The authors should make sure future readers don’t miss this.

This was done but somewhat hidden. It is now highlighted better (l. 163-166)

L44: “KOs” should be defined—better, it should be replaced with something that could be understood by someone not familiar with KOs.

Definition of KOs added in l. 44-46 and elaborated on the concept of gene variants thereafter.

L129: The sentence beginning with “Richness and diversity were independent...” isn’t clear. It’s not clear what is being correlated.

We modified this sentence for clarification:

L130-133: A correlation analysis of richness and diversity (see methods section) and read count showed no correlation in the taxonomic and gene profiles (Pearson correlation, 0.22 and 0.27, $p > 0.05$) but were significantly correlated to KO richness (0.78, $p \leq 0.01$).

L139: The variation in taxonomy and genes explained by the measured properties is very high, among the highest I remember seeing. Is that true, that it’s very high? Maybe worthwhile emphasizing more. Why were the authors able to see such as high percentage whereas others were not as successful?

We agree that the variation explained is truly very high. However, we are not aware of any other correlations of similar data sets analyzed by random forest models. Reviewer 3 made this valuable suggestion to use machine learning for such analyses. Therefore, it is hard or impossible to compare to other data. Rather, our analysis may set a new standard for such analyses in the future.

The high explained variation may also show that the latitudinal transect with systematically selected stations across the Atlantic Ocean from the (sub)antarctic to the northern temperate region provides a well suitable data set for such analyses. It may be an even better data set for such analyses than that of the stations from the Malaspina cruise which only visited tropical and subtropical stations and the Tara Ocean cruises with stations from very different biogeographic regions in different oceans.

L167: The sentence beginning here (“Our findings further imply that...”) is pure speculation without any connection to what the authors actually measured.

We disagree with this statement. Especially in the functional (gene & KO) datasets, we cannot explain the same amount of variance in the datasets by random forest modelling as for the taxonomical one. That indeed implies via reverse conclusion, that there are parameters we did/could not consider in our analysis that must be unknown drivers of functional properties.

L182: It doesn't make sense that nitrification has such a big impact on defining these clusters, because nitrifiers are much less abundant than heterotrophs and cyanobacteria. Perhaps the authors can find other data indicating nitrifiers are more abundant than would be expected.

We do not state, that nitrification is driving the observed clustering in our manuscript. The clustering is calculated from the complete KO dataset of which nitrification is only a minor part. However, in this publication we can only show a limited number of pathways that might be of interest for the community as stated in L176-178. Further analysis of a specific pathway would certainly distract from the core messages of this study i.e. functional differences in distinct clusters of the Southern/Atlantic Ocean.

L189: There is no need to use abbreviations here for the CAZymes.

Now I. 187: Since CAZyme identifiers use this terminology, we find it important to introduce the abbreviations in the text, as they reoccur in Figure 4 and later in the text.

L208: The Figure 4 discussed here is not the Figure 4 given in the paper. The discussion is about how CAZyme clusters varied with temperature, but the actual Figure 4 shows how dissimilarities in taxonomy, KOs, and genes varied with distance and temperature.

We apologize for this confusion. We changed the order of Figures and they are now in the proper order.

L210: There is no need to introduce the abbreviation "cC."

We removed these abbreviations.

L231: No need for "TD."

We removed these abbreviations.

L232: The text should cite the current Figure 4, not Figure 5.

We corrected the order of Figures (I. 230).

L242: The text following the sentence that starts here ("Recent findings of high gene variant...") is pure speculation not related to the authors' data. It should be deleted.

We relate our findings to data from the Mediterranean Sea and Tara Ocean. Hence the statement of this sentence is not speculative and supported by references 9 and 16.

L256: The text should cite the current Figure 5, not Figure 6.

Corrected. It is now Fig.

L277: The authors say that "different KOs vary in their temperature ranges and means" but then in

the next sentence, they admit the differences are not significant. They still say the trends are “obvious and worth exploring.” Maybe. The only thing “obvious” is the lack of any significant difference. That should be the conclusion, not that the KOs varied with temperature.

We deleted obvious in this sentence but want to keep this speculation, as pointed out above.

L279: Given the lack of a significant difference, the speculation about how different genes vary with temperature is not warranted and should be deleted.

As pointed out above, we want to keep this speculation but toned down our statement by deleting “obvious” from the text. However, since we did not claim any significance and only describe observations and we see no reason to delete this sentence. Since this is the first account of a detailed analysis of temperature dependencies and divergent adaptation in the marine environment we feel justified reporting on observed trends, even if they are not statistically significant, to motivate others to investigate the same in future studies to come to more conclusive results.

L285: This speculation should be deleted.

This is no speculation and supported by references 36-38.

Figure 2: The caption needs to explain how the lines were drawn and the meaning of the shading around each line. I think the shading should be removed—it only complicates trying to understand the graph.

We added the explanation to the Figure legend: “Shading indicates the 95% confidence interval of the regression.”

Figure 3: For the x-axis label, “difference” seems more appropriate than “change” because what is plotted is the difference between two clusters, not a change over time or space. Also, how can there be two, e.g. “4-fold”, for each I presume one is negative, but not sure how that is even possible. DEseq2 reports its results as “log2 fold change” and therefore our x-axis label is accurate. While calculating the log2 fold change, the operators of the results are depending on the order of comparisons made. Cluster I vs. Cluster II will give the same results as Cluster II vs. Cluster I but with flipped positive and negative values. We chose to use absolute values and indicate the “direction” of the enrichment in the Figure itself by colour code and position of the bar plots.

Figure 6: The caption for panel a needs to explain the r^2 values. I don’t see the need for panel b—it can be deleted or moved to supplemental materials. The tick labels for panels a and c should be rotated 90 degrees so readers don’t have to rotate their head to read them. The trend lines in panel c are too thick.

We adapted the Figure and moved the dendrogram to the supplement as Figure S7. We agree that it does not fundamentally help to understand the presented data.

The caption for panel c needs to explain why there are three patterns to how CAZyme abundance varies with temperature. These panels are really interesting.

The explanation was added in figure caption. “Relationships between temperature and CAZyme-family cluster abundance profiles determined by unimodal regression analysis.”

Reviewer #2 (Remarks to the Author):

The authors have done a nice effort integrating most of the requests and questions raised by the different reviewers and the manuscript has now a much better structure and flow. They have answered properly to all my concerns, deriving a new figure (Fig3) and I found their current version well improved and suitable for publication.

Thank you for this very positive remark

Reviewer #3 (Remarks to the Author):

In the submitted article, "Significance of gene variants for the functional biogeography of the sunlit Atlantic Ocean's microbiome," authors Dlugosch et al examine the environmental structuring of microbial taxonomic and functional biogeography across the Atlantic Ocean basin. Their results are transformative for the fields of biological oceanography and microbial ecology in that they convincingly demonstrate that, at high genomic resolution, functional genes show systematic and quantitative spatial patterns with both biogeographic province and temperature.

Previously, my main criticism was that I felt the authors interpretation of their results often lacked nuance or their results were not presented in a novel way. The authors have made a significant effort to refine their manuscript. As a result, the authors now present a more sophisticated analysis that is better contextualized in regards to the broader literature.

Moreover, I am very excited to see the authors' random forest analysis. Their conclusion that temperature better predicted taxonomic composition but that POC and chlorophyll better predicted functional profiles has very interesting implications for both the spatial and temporal structuring of microbial community composition and function.

Thus, the authors have addressed all my previous concerns. I have a few minor comments regarding grammar and clarity. Once these are addressed I have no further comments.

Thank you for your very favourable remarks and again for your suggestion to use machine learning for the correlative analyses of taxonomy and function analysing their biogeography. This really was an important addition to the study.

Minor comments:

Line 26-28: This sentence is a bit confusing. Perhaps split into two sentences, i.e., "A cluster analysis yielded three clusters of KOs but five clusters of genes differing in the abundance of genes involved in nutrient and energy acquisition. Gene profiles showed much higher distance-decay rates than KO and taxonomic profiles."

We split this sentence into two.

Line 29: Add comma- "observed patterns in the functional profiles, whereas temperature..."

Changed as requested.

Line 37-38: Move comma- “due to their high abundance and enormous taxonomic and functional diversity, attempts...”

Changed as requested.

Line 46: What is your exact definition of a “gene variant”? Ostensibly it is a gene associated with a specific function, but is it also associated with a specific taxon? And if so, what taxonomic “level” (i.e., 100% nucleotide similarity for a specific gene? 97% similarity?). Please clarify this term.

Thank you for this clarifying questions. We specified the difference between KOs and gene variants in line 44-50 as well as in the methods section I. 434. Gene variant definition referring to our analysis added in the results section I. 248-250.

Line 105: Figure S2 has some typos and figure panel labelling errors in the caption

Corrected.

Line 137-140: Typo- “we fit random forest (rf) models”

We corrected this past tense even though fitted seems to be correct as well.

Figure 2 caption: Typo- “determined by random forest models”

Typo corrected.

Although the captions for figures 4, 5, and 6 are correct, the order of the submitted images for these figures is shuffled

Order is corrected.

Reviewers' Comments:

Reviewer #1:

None